# MLL methyltransferases regulate H3K4 methylation to ensure CENP-A assembly at human centromeres

**Kausika Kumar Malik[1,2], Sreerama Chaitanya Sridhara[1], Kaisar Ahmad Lone[1,3], Payal Deepakbhai Katariya[1,2], Deepshika Pulimamidi[1], Shweta Tyagi[1]***

**1** Laboratory of Cell Cycle Regulation, Centre for DNA Fingerprinting and Diagnostics (CDFD), Uppal, Hyderabad, India, **2** Graduate Studies, Manipal Academy of Higher Education, Manipal, India, **3** Graduate Studies, Regional Centre for Biotechnology, Faridabad, India

\* shweta@cdfd.org.in

**Data Availability Statement:** All relevant data are within the paper and its Supporting Information files. The RNA-seq data are deposited in the NCBI GEO database (GSE231942). The FCS files

## Abstract

The active state of centromeres is epigenetically defined by the presence of CENP-A interspersed with histone H3 nucleosomes. While the importance of dimethylation of H3K4 for centromeric transcription has been highlighted in various studies, the identity of the enzyme (s) depositing these marks on the centromere is still unknown. The MLL (KMT2) family plays a crucial role in RNA polymerase II (Pol II)-mediated gene regulation by methylating H3K4. Here, we report that MLL methyltransferases regulate transcription of human centromeres. CRISPR-mediated down-regulation of MLL causes loss of H3K4me2, resulting in an altered epigenetic chromatin state of the centromeres. Intriguingly, our results reveal that loss of MLL, but not SETD1A, increases co-transcriptional R-loop formation, and Pol II accumulation at the centromeres. Finally, we report that the presence of MLL and SETD1A is crucial for kinetochore maintenance. Altogether, our data reveal a novel molecular framework where both the H3K4 methylation mark and the methyltransferases regulate stability and identity of the centromere.

## Introduction

Centromeres are specialized regions on chromosomes that form a scaffold of the kinetochore, a multi-protein complex that links chromosome to spindle microtubules to facilitate faithful chromosome segregation during mitosis; failure of this process leads to chromosomal structural and numerical abnormalities often seen in pathological conditions such as cancer [1]. Centromeres are organized into 2 broad regions, the inner "core" centromere region flanked by large outer peri-centromere. The centromere is characterized by repetitive α-satellite DNA sequences, which consist of approximately 171 bp monomers organized in tandem to form higher-order repeat (HOR) arrays that range from 2 to 5 Mb and are species and chromosome specific [2]. Although, the function of the centromere is highly conserved among the eukaryotes, the α-satellite DNA sequences are not evolutionary conserved [3]. In fact, centromeres pose an evolutionary conundrum as they are epigenetically defined by—centromeric protein

underlying the flow cytometry data can be found in the FlowRepository (FR-FCM-Z6AH).

**Funding:** This work was supported by the Wellcome Trust/DBT India Alliance Senior Fellowship to S.T.[IA/S/18/2/503981 http://www. indiaalliance.org/] and CDFD core funds. DBT SRF Award to K.K.M [DBT Ref No.DBT-JRF/13/AL/235/ 2457 Dtd.24/07/2013]. CSIR SRF Award to K.A.L [CSIR Ref No.09/724(0141)/2019-EMR-I Dtd.24/ 12/2019].CSIR SRF Award to P.D.K [CSIR Ref No.09/724(0136)/2019-EMR-I]. The funders had no role in study design, data collection and analysis, decision to publish, or preparation of the manuscript.

**Competing interests:** The authors have declared that no competing interests exist.

**Abbreviations:** CENP-A, centromeric protein A; CENP-C, centromeric protein C; cenRNA, centromere RNA; ChIP, chromatin immunoprecipitation; DAVID, Database for Annotation, Visualization and Integrated Discovery; DEG, differentially expressed gene; FBS, fetal bovine serum; GOA, Gene Ontology Annotation; GO, Gene Ontology; HAC, human artificial chromosome; HJURP, Holliday junction recognition protein; HMT, histone methyltransferases; HOR, higher-order repeat; lncRNA, long noncoding RNA; MLL, mixed lineage leukemia; PEI, polyethylenimine; qRT-PCR, quantitative real-time polymerase chain reaction; SD, standard deviation; SEM, standard error of the mean; SET, Su(var)3-9, Enhancer-of-zeste, Trithorax; TAD, transcription activation domain.

A (CENP-A)—a histone 3 variant, and not by the presence of α-satellite DNA. Interestingly, centromere chromatin (here on centrochromatin) constitutes CENP-A nucleosomes interspersed with histone 3 nucleosomes, bearing posttranslational modifications such as histone 3 lysine 4 dimethylation (H3K4me2), lysine 9 acetylation (H3K9ac), and lysine 36 dimethylation (H3K36me2) providing a unique chromatin state [4–8]. Although, initially thought to be transcriptionally silent, centrochromatin is now known to be transcribed by RNA polymerase II (RNA Pol II) and produces centromere RNA (cenRNA) transcripts [6,9–13]. Moreover, histone modification, centromere transcription, and the cenRNA are important for the centromere and kinetochore assembly and function [4,6,8,11,13–21]. For instance, cenRNAs physically interact with CENP-A, centromeric protein C (CENP-C), and Holliday junction recognition protein (HJURP) to efficiently recruit these proteins at the centromeres [11,15]. Furthermore, several reports suggest that RNA Pol II-mediated centromere transcription and cenRNA ensure loading of CENP-A at the centromeres in a cell cycle-specific manner [4,6,11,13,15]. The specialized nature of centrochromatin has led various groups to investigate the importance of histone modifications and transcription at the centromeres, and kinetochore maintenance [4,6,8,22–25]. Using synthetic human artificial chromosome (HAC), Earnshaw's group has shown that H3K4me2 mark is not only essential for cenRNA transcription, but also its removal resulted in rapid loss of transcription leading to impaired CENP-A incorporation and eventually, kinetochore instability [4,6]. While the importance of the H3K4me2 mark for centromeric stability has been revealed, the identity of the histone methyltransferases (HMT) depositing this mark on the centromere remains elusive.

In eukaryotes, the lysine methyltransferase 2 (KMT2, SET1, or MLL) family of proteins deposit the H3K4 methylation marks. In humans, there are 6 members in this family including MLL1 (MLL or mixed lineage leukemia protein), MLL2, MLL3, MLL4, SET Domain Containing 1A (SETD1A), and SETD1B. While SETD1A is a global H3K4 tri-methyltransferase, MLL1-4 displays locus-specific methylation activity [26–28]. All members of this family activate transcription through the Su(var)3-9, Enhancer-of-zeste, Trithorax (SET) domain that is responsible for the methyltransferase activity of these enzymes. In addition, some members like MLL and MLL2 also use the transcription activation domain (TAD) to promote transcription [29]. Different reports implicate MLL family members in the assembly of the transcription pre-initiation complex and recruitment of RNA Pol II to target genes [30]. In fact, H3K4 methylation has been proposed to be a prerequisite for recruitment of the basal transcription machinery and initiation of transcription for several mammalian gene targets [31,32]. However, how this machinery works in the context of an active intergenic chromatin state such as centromeres is still not clear.

MLL family members are involved in a wide variety of roles. However, the role of these proteins in mitosis is recently coming to light. Interestingly, all mitotic functions described so far for the members of this family are involved in averting chromosome mis-segregation and thus maintaining genomic integrity [33–37]. We have previously reported the localization of MLL and SETD1A on spindle apparatus and shown that MLL regulates proper chromosome alignment and segregation using protein–protein interactions [33]. Here, we show that most MLL family members have a role in regulating transcription of cenRNA. We report that endogenous MLL and SETD1A bind to human centromeres and regulate centromere transcription. Furthermore, using MLL knock-out cell lines, we reveal MLL as the "writer" for H3K4me2 and its crucial role in sustaining the unique epigenetic state of the human centromeres. Interestingly, removal of MLL but not SETD1A augments centromere R-loops (or co-transcriptional RNA: DNA hybrids) and RNA Pol II at centromeres. We also observe that loss of MLL and SETD1A adversely affects kinetochore maintenance as recruitment of CENP-B, CENP-C, and HJURP are compromised. Finally, we show that MLL and SETD1A affect the loading of nascent

CENP-A during the early G1 phase. Our results provide insights into centromere transcription and reveal a functional difference between the different members of the MLL family in regulating intergenic transcription.

## Results

### Members of the MLL family regulate centromeric transcription

Several studies have shown that centromeres are transcribed by RNA Pol II in a unique environment on chromatin decorated with histone modification like H3K4me2 and H3K36me2 [4,6,8,38]. However, how this transcription is regulated, is not fully understood. As members of the MLL family are responsible for depositing the H3K4me2 marks on the genome, we postulated that they regulate transcription of centromeric RNA (cenRNA). To test this hypothesis, we used previously characterized siRNAs to knock down various members of the MLL family [35] and studied the effect on α-satellite cenRNAs. These were detected by quantitative real-time polymerase chain reaction (qRT-PCR) using universal primer set from α-satellite DNA, sequences which are present on all centromeres [11,39]. We observed about 50% decrease in MLL family member transcripts with similar decrease in their protein levels, which resulted in a corresponding decrease in transcription of α-satellite arrays (Figs 1A, S1A–S1E). Several studies have reported that the use of RNA Pol II specific inhibitors reduce cenRNA transcripts [11,13,40]. We used these Pol II inhibitors as a positive control in our experiments and observed decreased transcription from the centromeres upon treatment with Triptolide, α-amanitin, and CDK9 inhibitor—LDC000067 hydrochloride (Fig 1B).

As α-satellite DNA sequences are also known to be present in peri-centromeres, in addition to the universal α-satellite primer, we used well-characterized primers from chromosome 17 specific α-satellite arrays (Fig 1C; [15]). Chromosome 17 contains 3 α-satellite arrays—D17Z1, D17Z1-B, and D17Z1-C—which vary in their sequence and size of their HORs [41]. These arrays are functionally distinct, producing active as well as inactive array transcripts [15]. HORs of both D17Z1 and D17Z1-B have been shown to form centromeres independently [42] while the status of D17Z1-C is not clear [43]. In addition, we analyzed genes that are co-regulated by RNA Pol II and MLLs like *APOL4*, *PAX3*, and *HBB*. Once we knocked down different MLL family members, we observed a reduction in all 3 array-specific transcripts from chromosome 17 (Figs 1D, 1E and S1F). Treatment with the 3 Pol II inhibitors, similarly reduced cenRNA transcripts from D17Z1, D17Z1-B, and D17Z1-C (S1G Fig). To sum up, our results indicate that all members of the MLL family tested here facilitate RNA Pol II-mediated transcription of cenRNA.

### MLL and SETD1A require the SET domain to regulate centromeric transcription

In order to understand how MLLs regulate cenRNA transcription, we choose MLL and SETD1A to study the process further because loss of both these proteins is known to produce chromosome-segregation defects [35]—defects, which are also caused as a consequence of the perturbed transcription at the centromere [4,6,11,15,17,23,24,44]. Reduced transcripts or protein levels of MLL and SETD1A resulted in a reduction of both α-satellite as well as array-specific transcripts from chromosome 17 (Figs 1D, 1E and S1A, S1B and S1D). In order to determine if this reduction was specific to MLL and SETD1A, we analyzed transcripts when siRNA-treated cells were complemented with full-length protein (Figs 1D, 1E and S1H). We also used SET-domain deleted MLL/SETD1A protein(s) to determine the role of the SET domain in cenRNA transcription. We utilized stable cell lines made in U-2OS cells for this

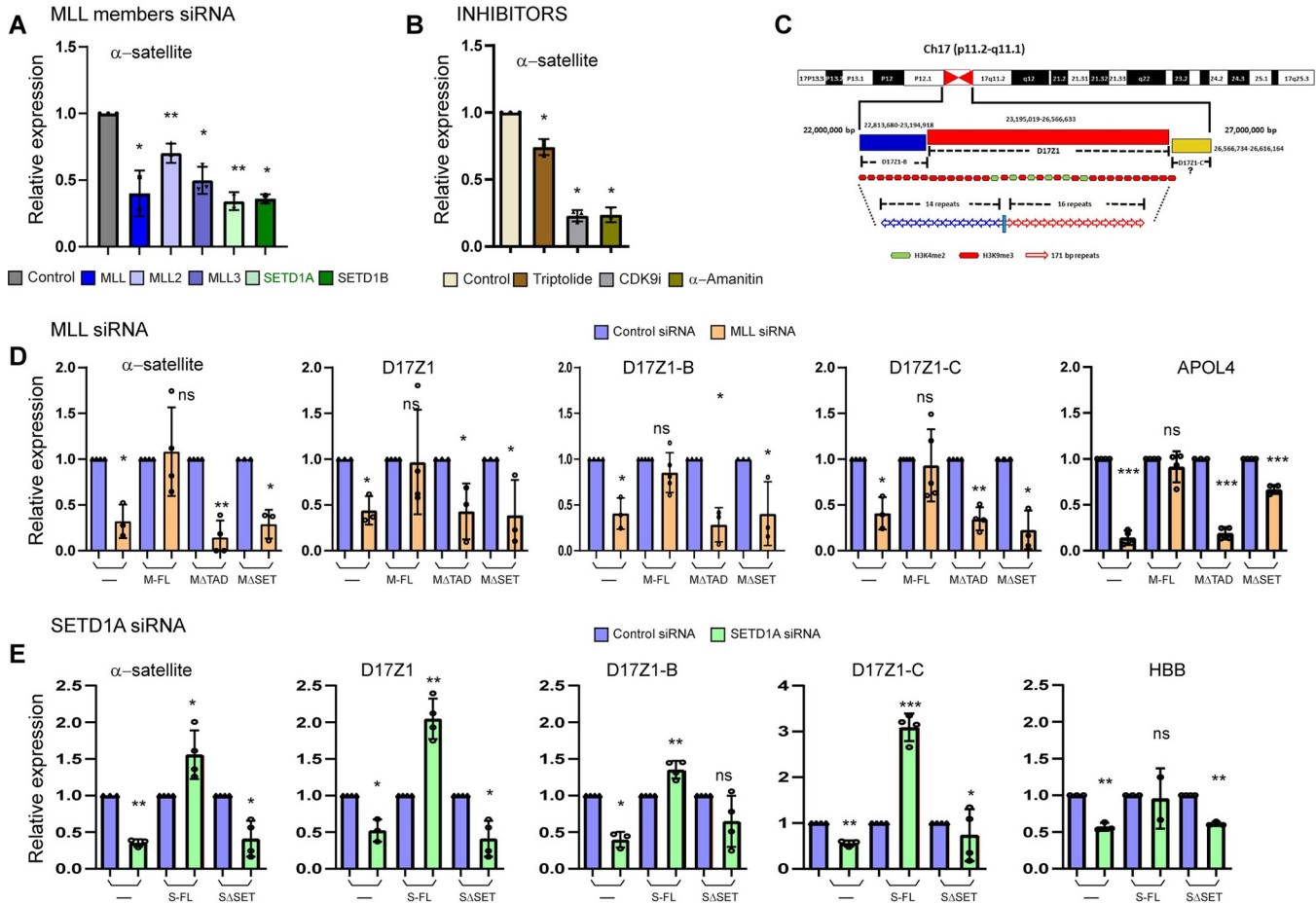

**Fig 1. RNAi-mediated down-regulation of MLL family members abrogates centromeric transcription.** (**A**) Shown is a qRT-PCR analysis of universal α-satellite cenRNA transcript level in Control, MLL, MLL2, MLL3, MLL4, SETD1A, and SETD1B siRNA-treated cells as indicated. (**B**) qRT-PCR analysis of α-satellite cenRNA expression after treatment with either Control (DMSO), Triptolide (20 μm), CDK9 inhibitor (20 μm), α-amanitin (20 μg), or for 4 h is shown. (**C**) Schematic representation of human chromosome 17 HOR α-satellite arrays—D17Z1, D17Z1-B, and D17Z1-C. The numbers indicate base pairs and are based on GRCh 38/hg38. (**D**) Following Control or MLL siRNA treatment, cenRNA transcript levels of α-satellite arrays as indicated were measured in parent U-2OS cells (—) or U-2OS cells stably expressing full-length MLL (M-FL), TAD deleted MLL (MΔTAD), and SET domain deleted MLL (MΔSET). (**E**) Shown is a qRT-PCR analysis of cenRNA transcripts as indicated, following treatment with Control or SETD1A siRNA in parent U-2OS cells (—) or U-2OS cells expressing full-length SETD1A (S-FL) and SET domain deleted SETD1A (SΔSET). cDNA was synthesized from total RNA after rigorous DNase treatment and amplified using qRT-PCR for indicated RNAs. Data from all samples were normalized to GAPDH mRNA levels from respective samples by using − ΔΔCT method and expression is shown relative to control siRNA-treated/DMSO-treated cells from respective cell line/treatment (which is arbitrarily set to 1). Data obtained for the α-satellite transcript in **A** from MLL and SETD1A siRNA treatment of parent U-2OS cells are replotted in **D** and **E**, respectively, for ease of comparison. Each experiment was performed at least 3 or more times except α-amanitin treatment (2 times). Error bars represent SD. *$P \leq 0.05$, **$P \leq 0.005$, ***$P \leq 0.0005$, ns: not significant, $P > 0.05$ (two-tailed Student's $t$ test). The raw data underlying parts (**A**, **B**) and (**D**, **E**) can be found in S1 Data. CDK9i, CDK9 inhibitor; cenRNA, centromere RNA; HOR, higher-order repeat; MLL, mixed lineage leukemia; qRT-PCR, quantitative real-time polymerase chain reaction; SD, standard deviation; TAD, transcription activation domain.

purpose (S1H Fig, [35]). To ensure that only endogenous *MLL* or *SETD1A* transcript is affected by our siRNA treatment, and not the recombinant one, we made use of siRNA directed against 3′ UTR of precursor mRNA (MLL siRNA #2) or made recombinant siRNA-resistant constructs by introducing silent mutations (S1I Fig, SETD1A siRNA#1). Our findings indicate that MLL and SETD1A specifically regulate centromeric transcription and this regulation is dependent on the SET domain. Interestingly, when we studied the TAD deletion in MLL (MLLΔTAD), we found that TAD also affects the transcript levels at the centromere (Figs 1D and S1H). Taken together, our findings suggest that MLL and SETD1A use their

transcription-competent domains to regulate cenRNA transcription from both active as well as inactive arrays.

## MLLs bind to the human centromere repeats

MLLs regulate the transcription of a large number of genes, either by direct binding to gene targets or indirectly [31,45–48]. Therefore, we wanted to investigate if MLL and SETD1A were present on the centromeres to regulate the cenRNA transcription. We checked for the presence of MLLs on the centromere using immunofluorescence staining (IF). Consistent with their role in the transcription of cenRNA, we found MLL and SETD1A co-localizing with CENP-A on the centromere in mitosis (Fig 2A). As cenRNA transcription has also been reported in early G1 cells, we performed IF in cells synchronized in this stage (Fig 2B). MLL and SETD1A showed co-localization with CENP-A not only in the early G1 phase but also in the asynchronous interphase cell population as well (Figs 2B and S2A). In contrast to the mitotic cells and consistent with their regulatory role in the genome, MLL and SETD1A localization on the centromere was distinct but not exclusive (Figs 2A, 2B and S2A). Further, when we modeled images of mitotic cells, we observed that even though there was a substantial overlap between MLL/SETD1A with CENP-A, the localization of these proteins was not restricted to CENP-A (S2B Fig). In order to further validate the specific signal of our proteins on the centromere, we performed siRNA-mediated depletion of MLL or SETD1A and observed reduced staining of these proteins, further confirming that MLL and SETD1A are specifically present on the centromeres (S2C and S2D Fig). This was accompanied by no primary antibody control (S2E Fig). We also checked for the presence of MLL2, MLL3, and SETD1B on the centromere (S2F Fig). Even though not as distinct as MLL or SETD1A, may be due to the fixation conditions or antibody used, these proteins were present on the centromeres (S2F Fig). Taken together, our data suggests that in addition to the canonical non-repetitive "regular" genomic loci, MLLs also bind to and regulate transcription from repetitive centromeric sequences transcribing noncoding RNA.

To confirm our observations from the IF, we performed chromatin immunoprecipitation (ChIP) using a specific antibody against MLL or SETD1A and checked for their binding on the centromeres in HEK-293 cells (Fig 2C and 2D). We used their canonical targets, i.e., *HOXA9* (& *PAX9*) for MLL and *RAD18* for SETD1A as positive control [31,49]. In our ChIP samples, we could detect enrichment of endogenous MLL and SETD1A protein over IgG on both α-satellite regions as well on chromosome 17 (Fig 2C and 2D). Independent of their centromere-forming status, we detected MLL and SETD1A on all 3 HORs in chromosome 17 (Fig 2C and 2D). In order to ascertain that the binding of MLL and SETD1A on centromeres is specific, we performed 2 additional experiments. First, we knocked down MLL (S2G Fig) or SETD1A (S2I Fig) using shRNAs, which enabled us to obtain sufficient cells for our ChIP assay. Consistent with the reduced binding of MLL and SETD1A on *HOXA9* and *RAD18* promoters, respectively, we observed that their enrichment was also significantly reduced on the α-satellite loci (S2H and S2J Fig). Second, we performed ChIP in non-transformed IMR90-tert cells and found that both MLL and SETD1A bound to the α-satellite region and chromosome 17 HORs in these cells as well (Fig 2E and 2F). We simultaneously performed ChIP with CENP-B, a DNA-binding protein that binds to a 17-bp consensus sequence present in α-satellite loci [50,51]. The CENP-B showed significant enrichment at centromeres but not at other non-centromeric loci (S2K Fig), indicating that we are able to amplify and detect centromere enrichment on endogenous chromosomes with our ChIP experiments. In addition to the H3K4 methyltransferase, we observed high levels of H3K4me2 in our ChIP experiments in both HEK-293 (Fig 2G) and IMR90-tert cells (Fig 2H). Altogether, our results show that both

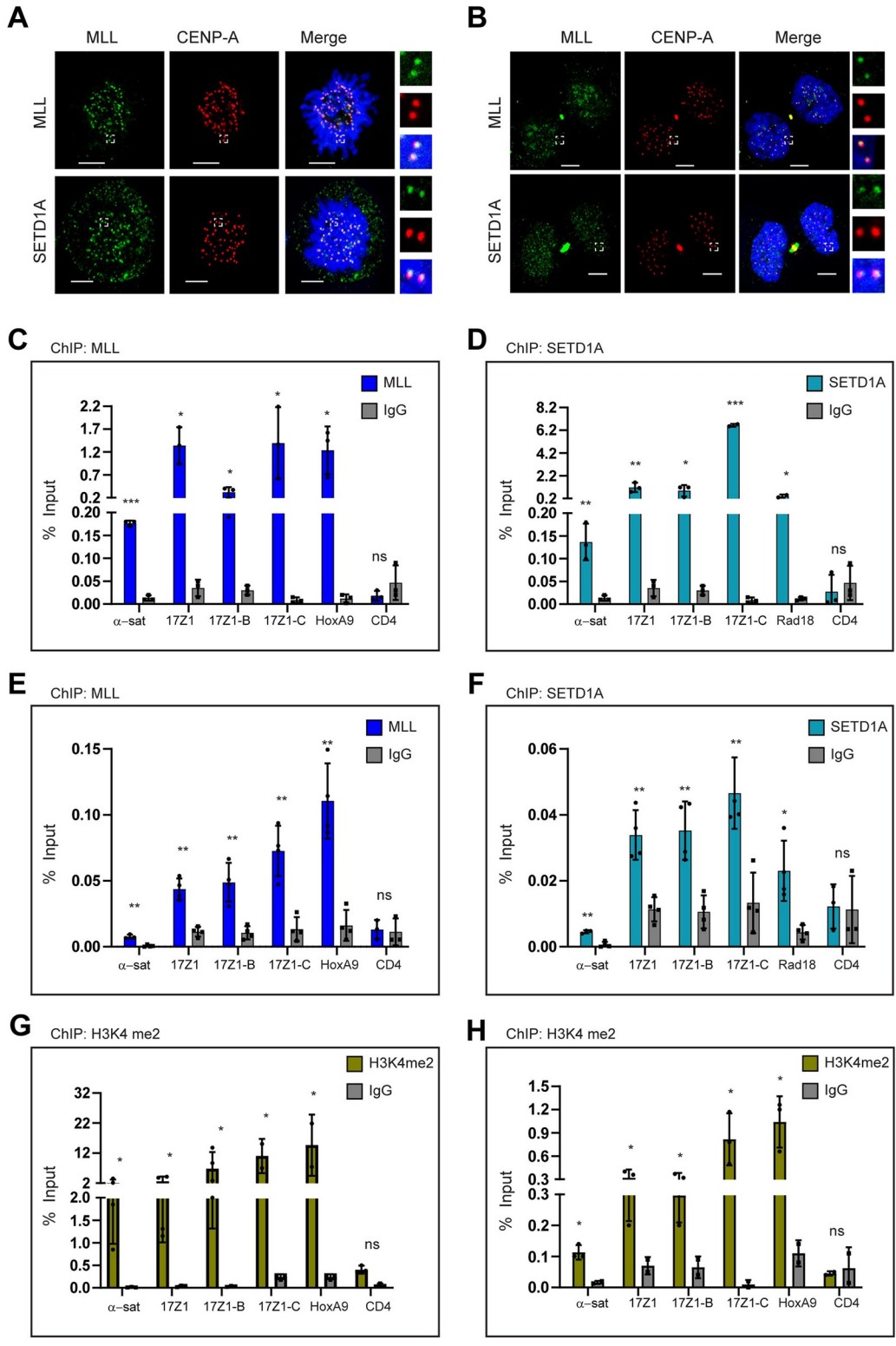

**Fig 2. MLLs binds to the human centromere repeats.** (**A**, **B**) Immunofluorescence staining (IF) of endogenous MLL (green) or SETD1A (green) with CENP-A (red) in U-2OS cells synchronized in mitosis (**A**) or early G1 (**B**) is shown. DNA was stained with DAPI (blue). The area in the white square is magnified and shown on the right for each image. Scale bar, 5 μm. Pearson correlation coefficient was measured for more than 100 centromeres and mean with SEM is shown between CENP-A and—MLL (**A**) = 0.45±0.015, SETD1A(**A**) = 0.50±0.014, MLL(**B**) = 0.35±0.015, and SETD1A

(**B**) = 0.50 ±0.013. (**C, D**) ChIP with MLL (**C**) or SETD1A (**D**) and IgG antibodies were performed on HEK-293 cells. Immunoprecipitated DNA was quantified with RT-qPCR and results plotted as percent input enrichment are shown. (**E, F**) Shown are analyzes of ChIP with MLL (**E**) or SETD1A (**F**), and IgG antibodies, performed on IMR-90 tert cells, as described above. (**G, H**) H3K4me2 and IgG chromatin immunoprecipitation was performed on HEK-293 (**G**) and IMR-90 tert (**H**) cells, and the result plotted as percent input enrichment are shown. Each experiment was performed at least 3 or more times. Error bars represent SD. *$P \leq 0.05$, **$P \leq 0.005$, ***$P \leq 0.0005$, ns: not significant, $P > 0.05$ (two-tailed Student's $t$ test). The raw data underlying parts (**C–H**) can be found in S1 Data. ChIP, chromatin immunoprecipitation; MLL, mixed lineage leukemia; α-sat, α-satellite; SD, standard deviation.

the H3K4 depositing enzymes as well as the H3K4 dimethylation marks are present at the centromeres.

## Loss of MLL affects the epigenetic landscape of the centromeres

Previous studies show that the H3K4me2 mark is essential for the transcription as well as the stability of the centromere [4,6,22]. Here, we have reported the presence of most members of H3K4 HMT family on the centromere (Figs 2 and S2F). In order to investigate, if these proteins indeed deposit the H3K4me2 marks on the centromeres, we decided to proceed with the analysis of one of these H3K4 HMT members—MLL—in greater detail. In the studies with HAC, it was observed that the effects of removal of the H3K4me2 mark were apparent after some days, as CENP-A turnover is slow [4,6]. Therefore, we decided to generate MLL knock-out cell lines. To achieve this, we performed CRISPR-Cas9 based genome editing on HEK-293 cells to produce MLL knock-out cell lines. Our initial attempts to generate MLL knock-outs in several different cell lines were not successful. Therefore, keeping in mind that MLL is essential for cell viability and growth [26,27], we generated and successfully obtained knock-outs in HEK-293 cell lines using doxycycline-inducible Cas-9 expression vectors ([52]; Fig 3A). MLL levels were drastically reduced in the 2 independent MLL-knock-out clones shown here (MLL iKO#11 and iKO#20) and both KOs showed cell cycle profile similar to Control cells (S3A Fig). In the case of chromatin-binding proteins, often cellular levels show reduction but not the chromatin-bound fraction. We, therefore, confirmed that MLL chromatin binding was indeed reduced in our inducible knock-out cell lines by ChIP assays (Figs 3B and S3B), both on centromere and *HOXA9* promoter. After successfully replicating these ChIP experiments in the iKO cell lines several times, we interrogated the effect of loss of MLL on H3K4 dimethylation levels. As expected, the H3K4me2 levels were dramatically reduced in both MLL iKO cell lines (Figs 3C and S3C), indicating that MLL was indeed one of the writers of H3K4me2 at the centromeres. Consistent with the observations on HAC, reduction in the H3K4me2 mark was accompanied by a reduction in H3K9 acetylation as well as an increase in H3K9me3 (Figs 3D, 3E and S3D and S3E). Surprisingly, despite reduced transcription upon loss of MLL, we did not observe a decrease in the level of the dimethylation at H3K36, rather it showed an overall increase (Figs 3F and S3F). These observations are in contrast with the results obtained in HAC, where the H3K36me2 mark was reduced upon lysine-specific demethylase1/2 (LSD1/2) targeting [4,6]. Notably, the H3K36me2 mark was only increased at the centromere in MLL iKO cells but not at the canonical locus—*PAX9* promoter (Figs 3F and S3F). All in all, these observations indicate that MLL regulates the local epigenetic landscape of centrochromatin by regulating the levels of H3K4 dimethylation marks.

## Disparate impact of MLL and SETD1A on centromeric R-loops

Recently, several studies have reported the presence of R-loops on the centromere, which have been shown to affect centromeric stability [53–56]. As MLL is involved in the transcription of cenRNA, we asked if loss of MLL would influence the status of R-loops on the centromere. To

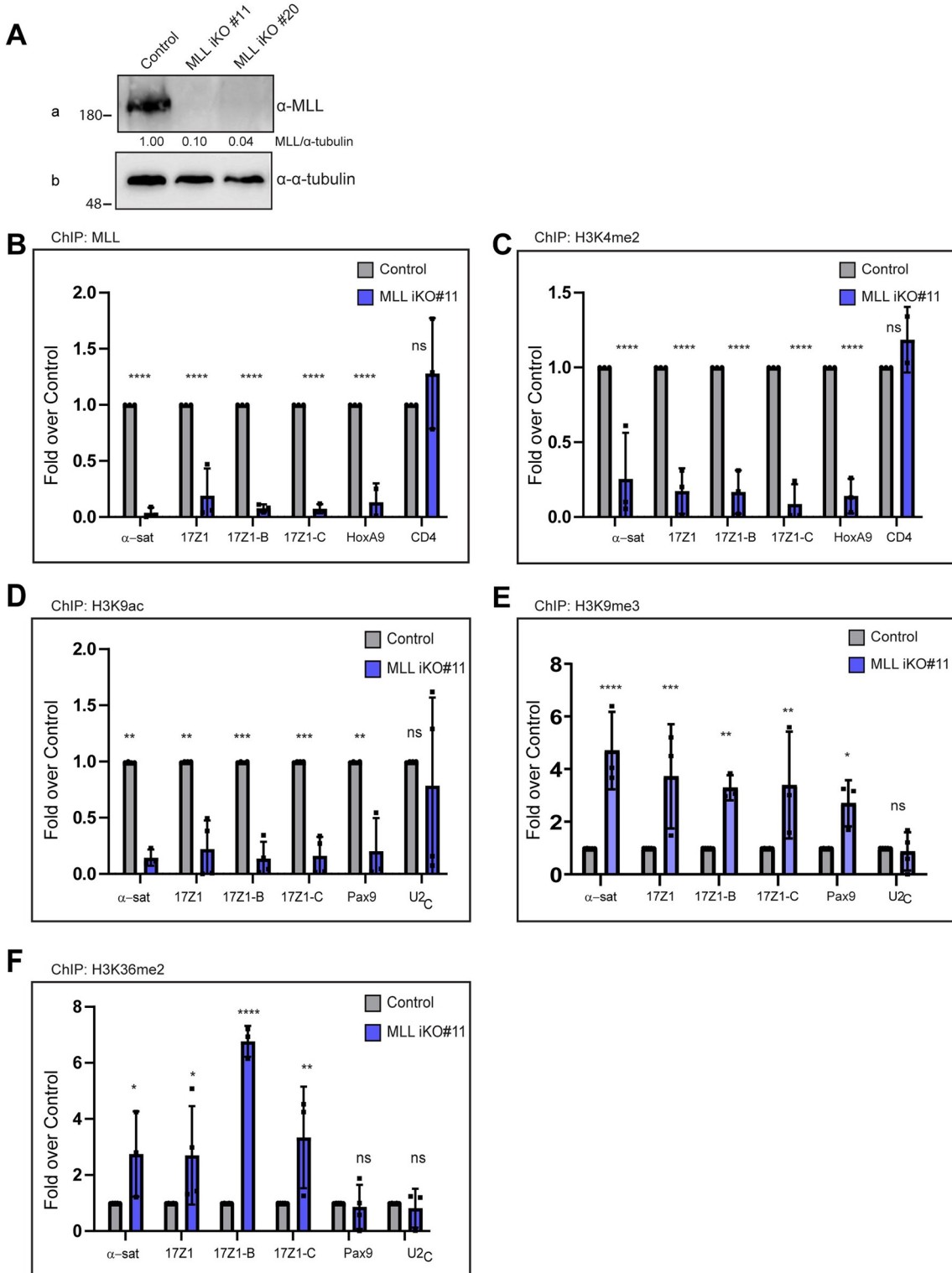

**Fig 3. Loss of MLL affects the epigenetic landscape of the centromeres.** (**A**) Immunoblot shows MLL protein levels in CRISPR-Cas9 generated inducible *MLL* knockouts (iKO) cells. Two independent clonal cell lines (#11 and #20) were used here. Blots were probed with α-MLL$_C$ and α-tubulin as shown (please note that the same sample was loaded on a different SDS-PAGE gel to evaluate tubulin). Molecular weight markers (in kDa) are shown on the left. Uncropped blots provided in S1 Raw Images. (**B–F**) ChIP analyzes using following antibodies: MLL (**B**), H3K4me2 (**C**), H3K9ac (**D**), H3K9me3 (**E**), and H3K36me2 (**F**) in *MLL* iKO cells (#11) are shown.

Data were normalized against the ChIP values obtained in parental (or Cas9-expressing) cells, which are used as Control. Each experiment was performed at least 3 or more times. Error bars represent SD. *$P \leq 0.05$, **$P \leq 0.005$, ***$P \leq 0.0005$, ****$P \leq 0.0001$, ns: not significant, $P > 0.05$ (two-way ANOVA with Šídák multiple comparison test). α-sat, α-satellite. The raw data underlying parts (**B–F**) can be found in S1 Data. ChIP, chromatin immunoprecipitation; MLL, mixed lineage leukemia; SD, standard deviation.

this end, we first used the S9.6 antibody to detect global nuclear R-loops by IF. The R-loop signal intensity was quantified by measuring the mean signal intensity in the nucleus in Control Vs Test cells. When quantified, to our surprise, we found that loss of MLL resulted in a higher R-loop signal compared to Control U-2OS cells (Figs 4A and S4A). Given MLL's role in cenRNA transcription, this result was unexpected. Further, a recent study reported the loss of R-loops upon SETD1A siRNA treatment [57]. Therefore, we quantified R-loop levels in SETD1A siRNA-treated cells. In contrast to MLL and consistent with the previous report, loss of SETD1A resulted in lower R-loop formation compared to Control cells (Fig 4A). We next knocked down both MLL and SETD1A and observed that the R-loop levels in these samples were similar to those observed in MLL siRNA-treated cells (Fig 4A, compare sample 2 and 3 with 4). RNA Pol II inhibitors are known to reduce R-loop formation due to inhibition of transcription [53]. Consistent with this hypothesis, the R-loop accumulation decreased in Triptolide-treated Vs non-Triptolide-treated Control siRNA samples (Fig 4A, compare sample 1 with 5). We found that the treatment with Triptolide also diminished R-loop formation in MLL siRNA-treated cells when compared with MLL siRNA non-Triptolide-treated samples (Fig 4A, compare sample 2 with 6) but these were still higher than Triptolide-treated Control siRNA samples (Fig 4A, compare sample 5 with 6). However, we observed no significant change between SETD1A-siRNA non-Triptolide Vs Triptolide-treated samples (Fig 4A, compare sample 3 with 7). Similarly, MLL-siRNA Triptolide Vs MLL +SETD1A Triptolide-treated samples (Fig 4A, compare sample 6 with 8) exhibited same levels of R-loops. Taken together, our results indicate that loss of MLL, but not SETD1A, promotes R-loop formation and this formation seems to be dependent largely on transcription.

To validate our findings of R-loops in IF, we performed DNA:RNA hybrid immunoprecipitation (DRIP) assay in MLL or SETD1A siRNA-treated cells. The RNase H treated genomic DNA before the DRIP assay worked as a control to ensure a specific signal [58]. An R-loop-prone locus–*RPL13A* (intron 7 and exon 8; [59]), acted as a positive control, while R-loop-free loci–*SNRPN* and *EGR1* [58,59], and MLL negative locus—U2$_C$ region [60] acted as negative controls (S4B Fig). Consistent with our IF data, loss of MLL increased R-loop signal by several folds on the centromere (Fig 4B), whereas SETD1A siRNA treatment resulted in a reduction of centromeric R-loop formation (Fig 4C). The DRIP signal was significantly reduced by pre-treatment with RNase H in all cases (Control, MLL, and SETD1A siRNA) indicating that we can reliably detect specific R-loops in our experiments. R-loops have been reported on centromeres in specific phases of the cell cycle [54,56]. We could detect significantly higher levels of R-loop accumulation on chromosome 17 in Control siRNA-treated mitotic cells compared to asynchronous cell population (S4C Fig). However, this mitosis specific R-loop enrichment was not observed in cells treated with MLL-siRNA (S4D Fig), indicating that the function of MLL in resolution of R-loops lies outside of mitosis. To sum up, our results show that MLL and SETD1A behave differently in regulating R-loop formation on the centromere.

## Loss of MLL perturbs RNA Pol II distribution at the human centromeres

Centromeres are known to be transcribed by RNA Pol II and R-loops are a by-product of transcription [2,61,62]. In order to understand, why R-loops are accumulating upon loss of MLL but not SETD1A, we stained for RNA Pol II on the centromere. We looked at total RNA Pol II

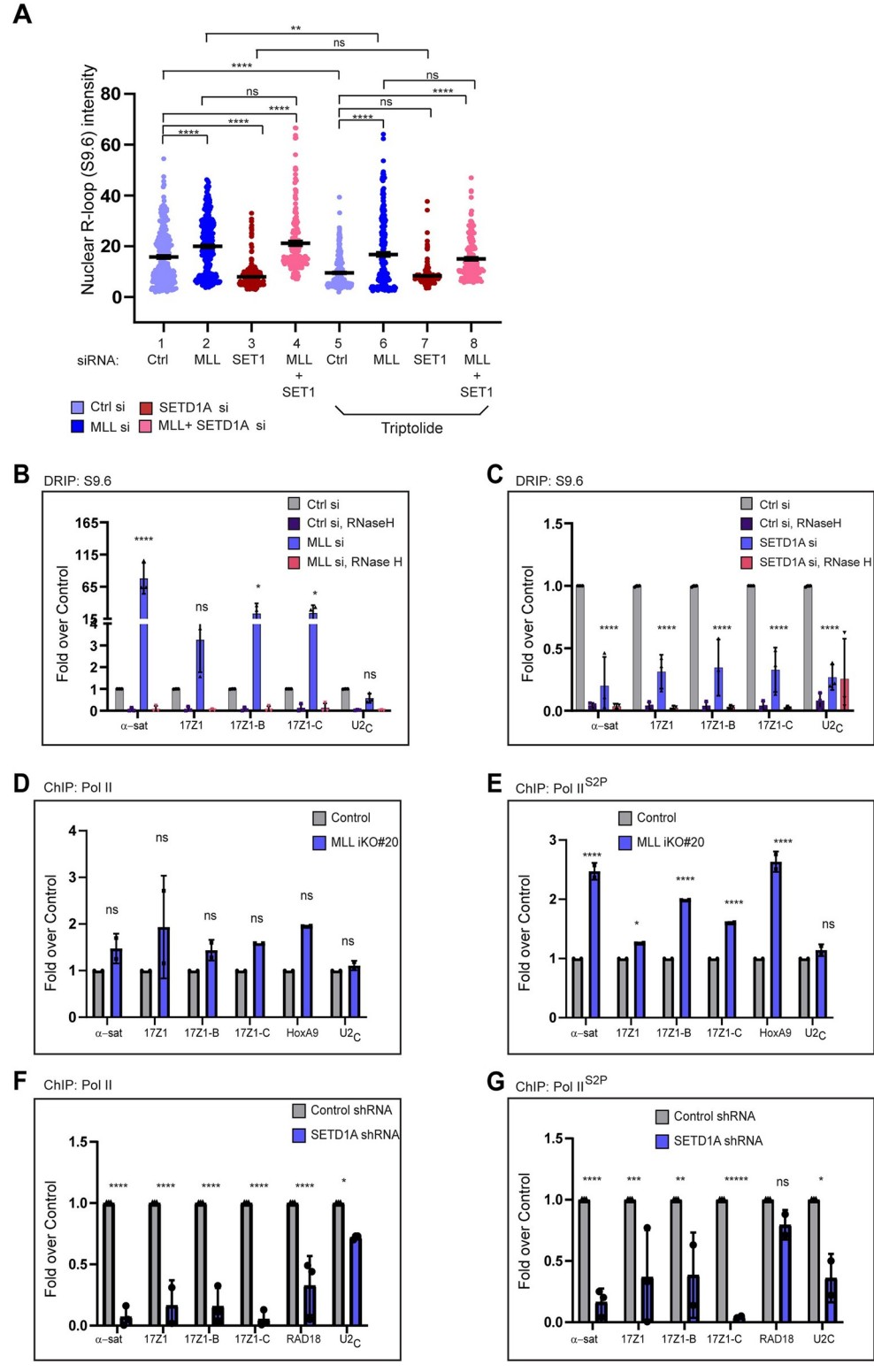

**Fig 4. Disparate impact of MLL and SETD1A on centromeric R-loops.** (**A**) Quantification of nuclear R-loops in U-2OS cells stained by S9.6 antibody, 48 h after Control, MLL, SETD1A, or MLL+SETD1A siRNA treatment, is shown. For transcription inhibition, cells were treated with 20 μm triptolide or DMSO (Control) for 4 h. The intensity of the whole nuclear R-loop staining is plotted. A total of 100 cells from 3 independent experiments was scored. See S4A Fig for representative images. Error bars represent SEM. ****$P \leq 0.0001$, **$P \leq 0.005$, ns: not significant, $P > 0.05$

(Ordinary one-way ANOVA with Tukey's multiple comparisons test). (**B**, **C**) DNA:RNA immunoprecipitations (DRIP) after MLL (**B**) and SETD1A (**C**) RNAi in HEK-293 cells, with respective RNase H controls, is shown. Data were normalized against the DRIP values obtained in Control siRNA-treated cells (Ctrl si). Note that the *U2C* control (in B) is the same data from S4B Fig but normalized against DRIP values obtained in Control cells. Each experiment was performed at least 3 or more times. Error bars represent SD. *$P \leq 0.05$, ****$P \leq 0.0001$, ns: not significant, $P > 0.05$ (two-way ANOVA with Šídák multiple comparison test). (**D**, **E**) ChIP-analysis of total RNA Pol II (**D**) and RNA Pol II$^{S2P}$ (**E**) in *MLL* iKO #20 cells are shown. Data were normalized against the ChIP values obtained in parental (or Cas9-expressing) cells, which are used as Control. Data from 2 independent ChIP experiments are plotted. (**F**, **G**) ChIP analysis of total RNA Pol II (**F**) and RNA Pol II$^{S2P}$ (**G**) in SETD1A shRNA-treated cells are shown. Data were normalized against the ChIP values obtained in Control shRNA-treated cells. Data from 3 independent ChIP experiments are plotted. (**D–G**) Error bars represent SD. *$P \leq 0.05$, **$P \leq 0.005$, ***$P \leq 0.0005$, ****$P \leq 0.0001$, ns: not significant, $P > 0.05$ (two-way ANOVA with Šídák multiple comparison test). The raw data underlying parts (**A–G**) can be found in S1 Data. ChIP, chromatin immunoprecipitation; Ctrl, control; MLL, mixed lineage leukemia; SD, standard deviation; SEM, standard error of the mean; SET1, SETD1A; sh, shRNA; si, siRNA;.

(S4E Fig) and elongating RNA Pol II as scored by RNA Pol II phosphorylated on CTD serine 2 (RNA Pol II$^{S2P}$) (S4F Fig), via IF staining on centromere after treating the cells with MLL or SETD1A siRNAs. We observed that there was little effect of MLL siRNA on the presence of both forms of RNA Pol II. In contrast to these observations and consistent with the model generated by HAC, where H3K4me2 facilitates RNA Pol II-mediated transcription, loss of SETD1A exhibited dispersed foci for both forms of RNA Pol II and, even displayed reduced intensity for RNA Pol II$^{S2P}$ (S4E and S4F Fig). Previous reports indicate that loss of MLL may result in abnormal distribution of Pol II at a subset of genomic loci [31,46,63]. In order to understand what is happening at the centromeres, we performed ChIP with total RNA Pol II and RNA Pol II$^{S2P}$ in MLL iKO cells. Our analysis revealed that the levels of total RNA Pol II did not show any significant change in Control Vs MLL iKO samples (Figs 4D and S4G). However, we did find increased levels of RNA Pol II$^{S2P}$ accumulated on the centromere upon loss of MLL (Figs 4E and S4H). In direct contrast, SETD1A-depleted cells exhibited reduced levels of both total RNA Pol II and RNA Pol II$^{S2P}$ (Fig 4F and 4G). Taken together, our results show that although MLL and SETD1A deposit H3K4me2 at the centromeres (Figs 3C and S4I; [57]), they differentially regulate RNA Pol II and thus R-loop resolution, at least at centromeres.

## MLL and SETD1A affect kinetochore maintenance and the recruitment of CENP-B and CENP-C to the centromere

Centromere transcription has been implicated in kinetochore maintenance as cenRNA is required for accurate localization of many centromere-associated proteins including CENP-C [15,24]. Recently, CENP-B was also shown to be bound by transcripts from inactive arrays [15,50]. As both MLL and SETD1A affect the transcription of cenRNA from active as well as inactive arrays, we analyzed the effect of MLL or SETD1A knockdown on the kinetochore maintenance. When we checked the localization of CENP-C and CENP-B on the centromere by IF, we observed that the centromeric levels of both CENP-C and CENP-B proteins were decreased upon MLL or SETD1A siRNA treatment (Fig 5A and 5B). We further analyzed these protein levels on the centromere in cell lines expressing different domain deletion of MLL or SETD1A upon siRNA treatment, and as shown, we observed a consistent decrease in both CENP-C (Fig 5A–5D) and CENP-B (Fig 5A, 5B, 5E and 5F) on the centromere. We observed an increase in the levels of CENP-C (and CENP-B) in cell lines expressing full-length MLL or SET1A protein (Fig 5C–5F). This can be partly explained due to an increase in cenRNA transcript observed upon expression of SETD1A full-length protein (Fig 1E), indicating that centromeric transcripts indeed play a role in recruiting/stabilizing CENP-C (and CENP-B) to the centromeres.

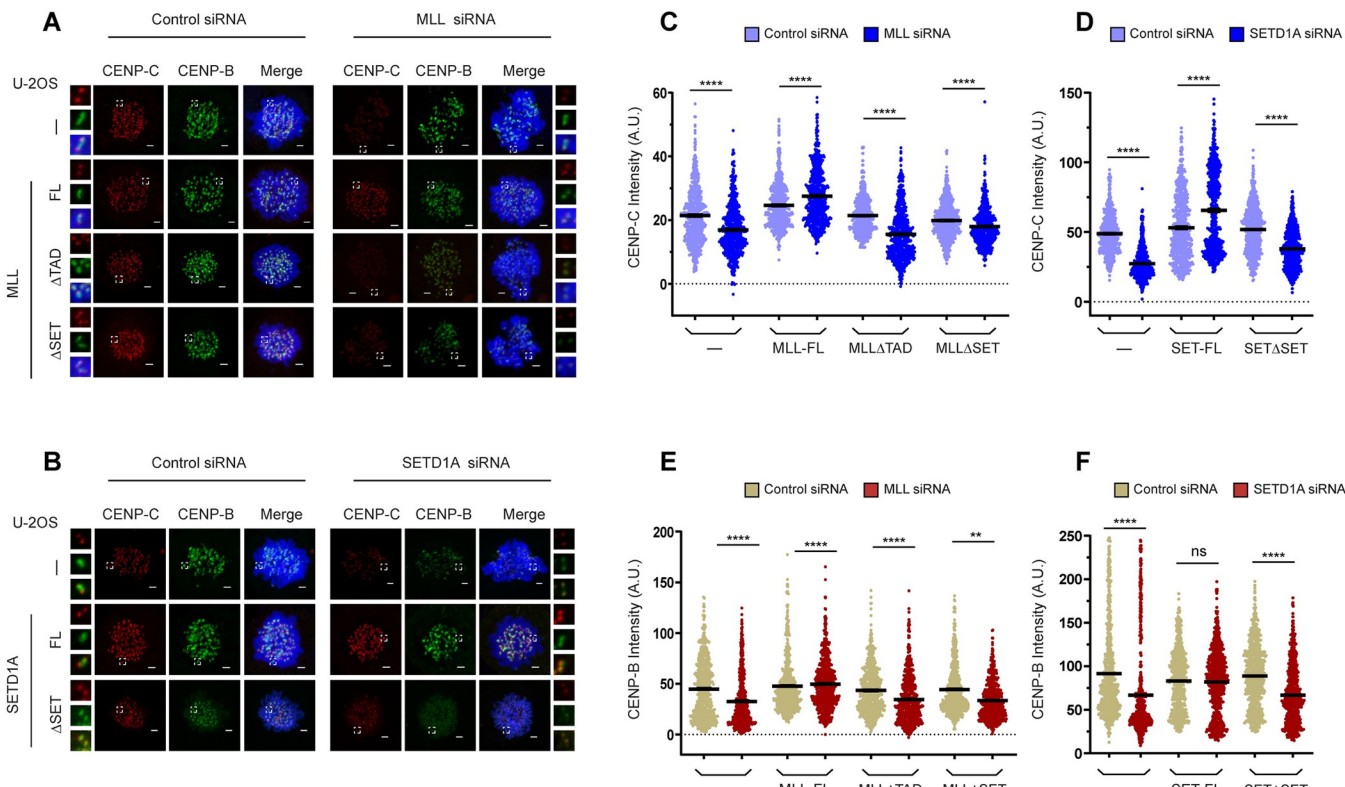

**Fig 5. MLL and SETD1A affect kinetochore maintenance.** (**A**) Representative IF images show mitotic CENP-C (red) and CENP-B (green) staining in parent U-2OS cells (—) or U-2OS cells expressing full-length MLL (MLL-FL), TAD deleted MLL (MLLΔTAD) and SET domain deleted MLL (MLLΔSET) following treatment with either Control or MLL siRNA. (**B**) Representative IF images show mitotic CENP-C (red) and CENP-B (green) staining following treatment with Control or SETD1A siRNA in parent U-2OS cells (—) or U-2OS cells expressing full-length SETD1A (SET-FL), and SETD1AΔSET (SETΔSET; here N1646A mutant was used). (**A, B**) DNA was stained with DAPI (blue). The area in the white square is magnified and shown on the side of each image. Scale bar, 2 μm. (**C–E**) The graph represents the quantification of CENP-C (**C**) and CENP-B (**E**) intensity in MLL depleted cells shown in **A**. (**D–F**) Quantification of CENP-C and CENP-B intensity following treatment with SETD1A siRNA as shown in **B**. (**C–F**) Each data point represents a single centromere. The error bar represents SEM $n \geq 300$ centromeres ($n = 2$ experiments). For quantification, Z-stack images were merged and individual CENP-C and CENP-B signal intensity were measured using ZEN software. ****$P \leq 0.0001$, **$P \leq 0.005$, ns: not significant, $P > 0.05$ (Mann–Whitney two-tailed unpaired test). The raw data underlying parts (**C, D**) and (**E, F**) can be found in S1 Data. MLL, mixed lineage leukemia; SEM, standard error of the mean; TAD, transcription activation domain.

## Identifying differentially regulated genes upon MLL and SETD1A knockdown

As loss of both MLL and SETD1A is likely to perturb the transcription of a large number of genes, and may impact the conclusions being drawn here, we decided to examine the gene expression analysis upon the knock down of these proteins in U-2OS cells. RNA-sequencing (RNA-seq) was performed 72 h after siRNA treatment and differential gene expression (DGE) signatures of Control Vs MLL KD, and Control Vs SETD1A KD were examined (S5 Fig). MLL depletion resulted in significant differential regulation of approximately 2,200 genes while in samples depleted of SETD1A, approximately 2,400 genes exhibited change in expression (S1 Table and S5A and S5B Fig). Gene Ontology (GO) enrichment analysis showed that MLL and SETD1A are primarily involved in signal transduction, cell adhesion and extracellular matrix organization in U-2OS cells as these biological processes were common among the top 10 enriched in both gene sets (S5C and S5D Fig). Interestingly, we identified only about 3% genes with GO term "cell cycle" and less than 1% genes with GO term "RNA pol II transcription" to be differentially expressed in MLL-KD and SETD1A-KD (S5E and S5F Fig and S2 and S3 Tables). Relevant to the study being undertaken here, we checked for the overlap of DEGs with

GO term "centromere" and found that loss of MLL did not show significant perturbation in any centromere genes (S5E Fig) while SETD1A showed down-regulation of 1 centromere gene (see below). However, we still went ahead and analyzed 15 gene transcripts including those of Constitutive Centromere Associated Network (CCAN) proteins, Mis18 complex, and others implicated in CENP-A recruitment (S4 Table). Reflecting our RNA-seq DEG analysis, only CENP-H showed down-regulation in SETD1A-KD (S2 and S4 Tables and S5G Fig). CENP-H has been implicated in the recruitment of newly synthesized but not the endogenous CENP-A [64]. However, recent work reports that the recruitment of CENP-H is downstream to the recruitment of CENP-C [65], and CENP-C is able to localize to the centromere in absence of CENP-H [66–69]. It is likely that the recruitment of CENP-C by SETD1A may still be an independent event. Nonetheless, CENP-H acts as a bridge between centromeric nucleosomes and kinetochore microtubules, and its down-regulation needs to be recognized while assessing the role of SETD1A in kinetochore maintenance.

### Regulatory role of MLL in kinetochore maintenance in *cis* and in *trans*

Even though transcript levels of all proteins being analyzed here (in Figs 5–7) remain unchanged upon loss of our HMTs, we decided to confirm this by analyzing their protein levels upon RNAi treatment of MLL and SETD1A. Consistent with our RNA-seq experiments, we found that the protein levels of CENP-C, CENP-B, HJURP, and CENP-A were unchanged in U-2OS cells upon treatment with MLL or SETD1A siRNA (S6A–S6H Fig). However, we did observe decreased levels of CENP-B but no other proteins in MLL iKO cells (S6I–S6L Fig). Further analysis using previously published data sets [63] indicated that CENP-B may be a direct transcriptional target of MLL in HEK-293 cell line.

While this paper was in revision, Zhang and colleagues reported that a long noncoding RNA (lncRNA) named CENP-C targeting transcript (CCTT), interacts with CENP-C through specific RNA–protein interactions and regulates CENP-C level at centromeres [70]. As MLLs are involved in regulating the transcription of lncRNA [71,72], we tested the levels of CCTT in MLL and SETD1A depleted cells. Our transcript analysis showed that CCTT levels showed a significant decrease upon SETD1A but not MLL depletion (S5H Fig). Taken together, our results indicate that MLL family regulates the kinetochore maintenance at several levels—in *cis* by modulating transcription of cenRNAs at centromeres or CCTT at Chromosome 17, and therefore impacting the recruitment of centromere proteins (like CENP-C) or in *trans* by regulating transcription of centromeric/kinetochore gene (like *CENP-B* and *CENP-H*). All in all, loss of MLL and SETD1A have an adverse effect on the maintenance of kinetochores.

### Loss of MLL and SETD1A affects recruitment of CENPA at centromeres

Both CENP-C and CENP-B participate at different levels to stabilize CENP-A nucleosomes [73–76]. Further, the recruitment of CENP-A chaperone HJURP is dependent on H3K4me2-facilitated transcription [6]. Hence, we sought to determine if the recruitment of HJURP on the centromere, and as a consequence that of CENP-A, is affected upon loss of our H3K4 HMTs. We depleted MLL or SETD1A in our parent U-2OS cell line and co-stained the cells for endogenous HJURP and CENP-A in early G1 cells (Fig 6A and 6B). Consistent with results observed with CENP-C, HJURP and CENP-A recruitment was diminished in parent U-2OS cells as well as cells expressing MLLΔTAD and MLLΔSET treated with MLL siRNA, but not full-length MLL (Fig 6A, 6C and 6E). Similarly, upon treatment with SETD1A siRNA, parent U-2OS cells and SETD1AΔSET cell line showed significant loss of HJURP and CENP-A on the centromere, while expression of full-length SETD1A was able to restore their levels (Fig 6B, 6D and 6F). When MLL and SETD1A were depleted simultaneously, cells showed further loss

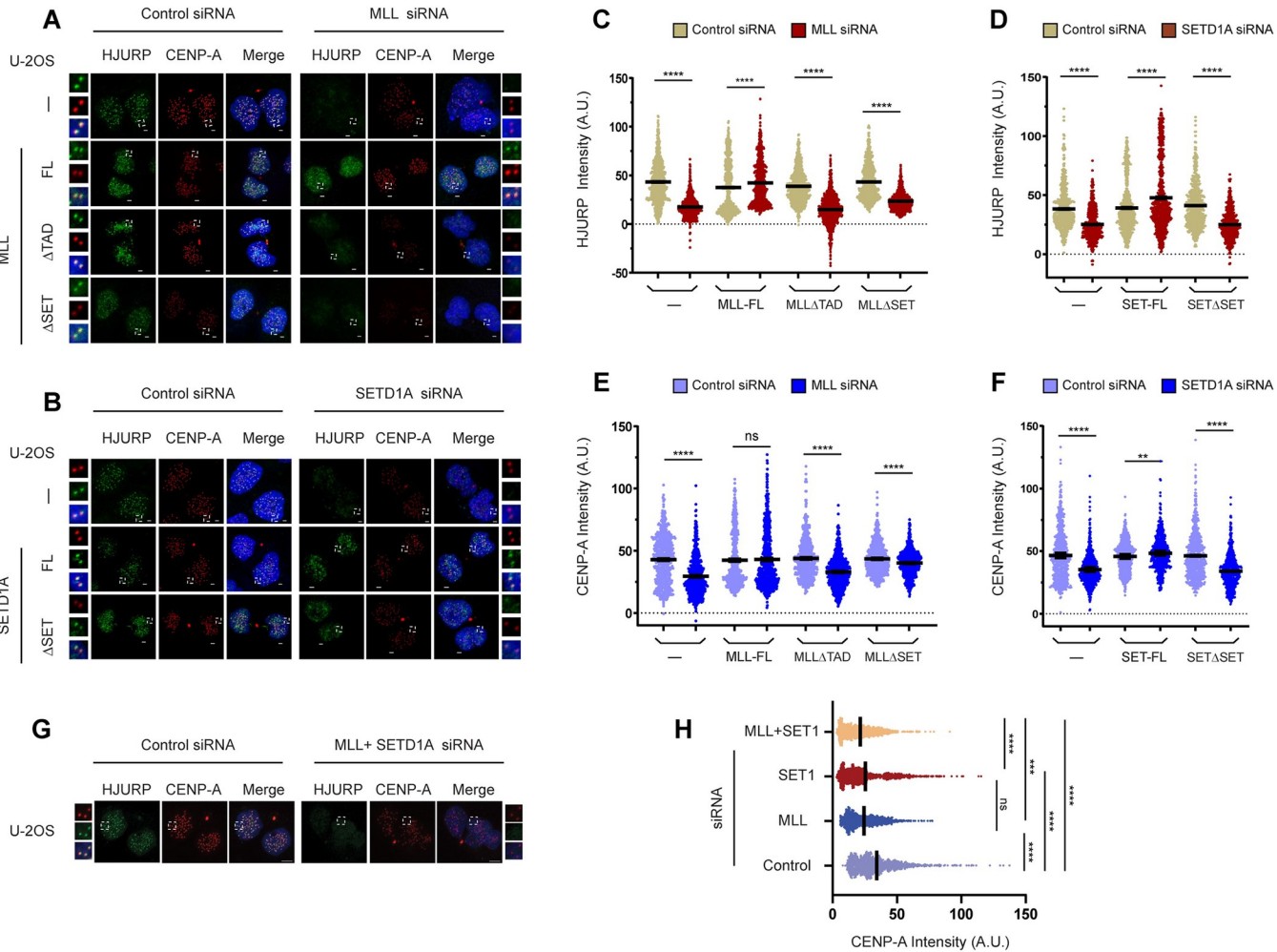

**Fig 6. Loss of MLL and SETD1A affects recruitment of CENPA at centromeres.** (**A**) The representative IF images show the loading of HJURP and CENP-A at the centromere in early G1 phase cells. Parent U-2OS cells (—) or U-2OS cells expressing recombinant full-length MLL (MLL-FL), TAD deleted MLL (MLLΔTAD), and SET domain deleted MLL (MLLΔSET), were treated with either Control or MLL siRNA. Cells, synchronized in the early G1 phase, were stained with α-CENP-A (red) and α-HJURP (green) antibody as shown. (**B**) Parent U-2OS cells (—) or U-2OS cells stably expressing recombinant full-length SETD1A (SET-FL), and SET domain deleted SETD1A (SETΔSET) were treated with either Control or SETD1A siRNA. (**A, B**) DNA was stained with DAPI (blue). The area in the white square is magnified and shown on the side of each image. Scale bar, 2 μm. (**C–F**) Quantification of HJURP and CENP-A fluorescence intensity following depletion of MLL (**C, E**) and SETD1A (**D, F**) respectively. ****$P \leq 0.0001$, **$P \leq 0.005$, ns: not significant, $p > 0.05$ (Mann–Whitney two-tailed unpaired test). (**G**) Parent U-2OS cells were treated with either Control or MLL+ SETD1A combined siRNA. Cells, synchronized in the early G1 phase, were stained with α-CENP-A (red) and α-HJURP (green) antibody as shown. Scale bar, 5 μm. (**H**) Quantification of CENP-A fluorescence intensity following depletion of Control, MLL, SETD1A, or MLL+ SETD1A. ****$P \leq 0.0001$, ***$P \leq 0.001$, ns: not significant, $p > 0.05$ (Ordinary one-way ANOVA with Tukey's multiple comparisons test). Quantification of HJURP intensity for **G** is shown in S7A Fig. (**C–F, H**) Each data point represents a single centromere. The error bar represents SEM ≥250 centromeres quantified from 10 early G1 cell pairs, ($n$ = 2 experiments). The raw data underlying parts (**C, D**) and (**E, F, H**) can be found in S1 Data. A.U., arbitrary units; MLL, mixed lineage leukemia; SEM, standard error of the mean; SET1, SETD1A; TAD, transcription activation domain.

of HJURP and CENPA compared with cells depleted of either MLL or SETD1A alone (Figs 6A, 6B, 6G and 6H and S7A). Our observations indicate that both MLL and SETD1A contribute additively to CENP-A recruitment.

## MLL and SETD1A facilitate recruitment of nascent CENPA at centromeres

In order to confirm that the loss of CENP-A observed here is not due to reduced cell proliferation upon loss of MLL or SETD1A [35,77,78], we made use of the pulse-chase labeling

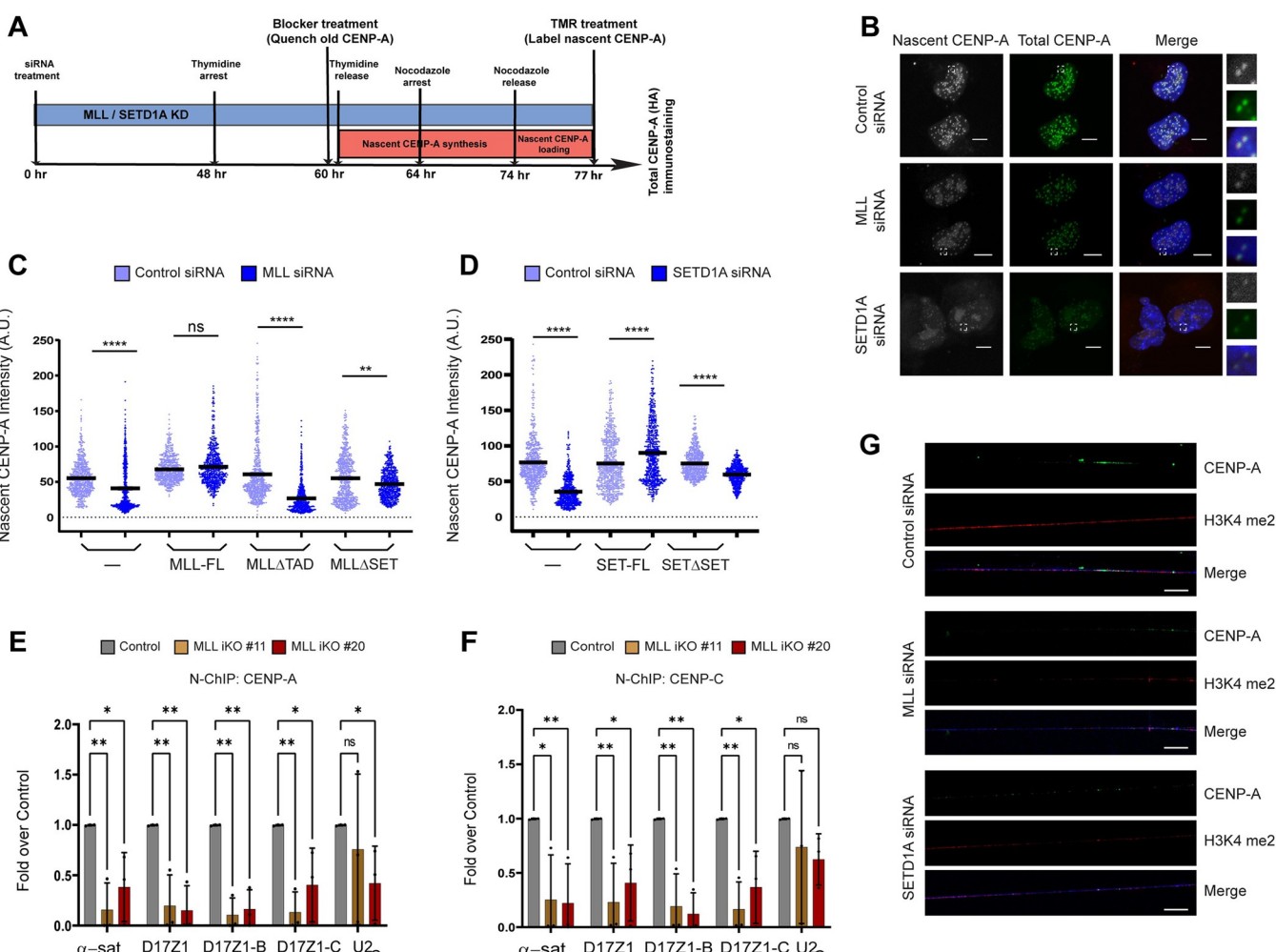

**Fig 7. MLL and SETD1A facilitate recruitment of nascent CENPA at centromeres.** (**A**) Schematic of cell synchronization and TMR-based labeling strategy to detect nascent CENP-A upon MLL/SETD1A knockdown is shown. See methods for more details. (**B**) Representative IF images showing the effect of Control/MLL/SETD1A siRNA treatment on nascent CENP-A loading at the centromere in U-2OS cells are shown. Cells stably expressing the CENP-A-SNAP-HA construct were used for siRNA treatment. Nascent CENP-A was labeled with TMR (gray) while total CENP-A was detected by IF using α-HA antibody (green). The area in the white square is magnified and shown on the left for each image. Scale bar, 5 μm. (**C**) Quantification of centromeric fluorescence intensity of nascent CENP-A SNAP in parent U-2OS cells (—) or cell line stably expressing siRNA resistant MLL full length (FL) or MLLΔTAD or MLLΔSET upon MLL siRNA. (**D**) Quantification of centromeric fluorescence intensity of nascent CENP-A-SNAP in parent U-2OS cells (—) or cell line stably expressing siRNA-resistant SETD1A full length (FL) or SETD1AΔSET (here, N1646A mutant was used) upon SETD1A siRNA is shown. (**C, D**) Each data point represents a single centromere; $n \geq 300$ quantified from 10 early G1 cell pairs ($n = 2$ experiments). ****$P \leq 0.0001$, **$P \leq 0.005$, ns: not significant, $p > 0.05$ (Mann–Whitney two-tailed unpaired test). (**E, F**) Shown are native ChIP analysis of CENP-A (**E**) and CENP-C (**F**) in *MLL* iKO cells (#11 and #20). Data were normalized against the ChIP values obtained in parental (or Cas9-expressing) cells, which are used as Control. Data from 3 or more independent ChIP experiments are plotted. Error bars represent SD. *$P \leq 0.05$, **$P \leq 0.005$, ns: not significant, $P > 0.05$ (two-way ANOVA with Šídák multiple comparison test). (**G**) Chromatin fibers were prepared by transfecting either Control, MLL, or SETD1A siRNA in parent U-2OS cells, followed by extraction of chromatin fibers and staining with endogenous CENP-A (green) and H3K4me2 (red), and DNA stained with DAPI (blue). Frequency of H3K4me2 co-localizing with CENP-A was found to be 17/19 in Control. This dropped to 5/21 upon MLL siRNA treatment and 0/19 upon SETD1A siRNA treatment. Scale bar, 10 μm. The raw data underlying parts (**C, D**) and (**E, F**) can be found in S1 Data. A.U., arbitrary units; ChIP, chromatin immunoprecipitation; MLL, mixed lineage leukemia; SD, standard deviation.

approach by using SNAP tagged CENPA [79] that can reveal if incorporation of newly synthesized CENP-A is affected or not. We have shown that treatment with MLL or SETD1A siRNAs resulted in a 50% reduction in cenRNA after 72 h (Fig 1A). At this time, despite MLL KD, at least 50% of cells are still division-competent [35]. Therefore, 60 h after siRNA treatment, the old CENP-A was blocked and newly synthesized CENP-A loading was detected by TMR-

staining in cells synchronized in early G1 phase (Fig 7A). As shown, we observed about 50% to 60% reduction in nascent CENP-A loading upon perturbing MLL or SETD1A levels (Fig 7B–7D). Interestingly, this decrease was restricted to 15% to 25% upon compromising the HMT domains of MLL or SETD1A (Fig 7C and 7D). On the other hand, TAD deletion in MLL caused about 42% decrease in nascent CENP-A levels, much nearer to the levels observed in parent U-2OS cells (Fig 7C). Our findings suggest that inhibition of potent transcription, like by the TAD, shows a more immediate effect on nascent CENP-A loading than depletion of H3K4me2, whose effects may be manifested over a longer period of time. Remarkably, we also observed a decrease in the exogenously expressed total CENP-A levels on the centromere when their protein levels were unchanged (or increased; S7B–S7E Fig). Our results indicate that MLL or SETD1A depletion was more deleterious than RNAi-mediated depletion of cen-RNA [11,15] or LSD1/2 -mediated removal of H3K4me2 [4,6]. Encouraged by our observations here, we performed native ChIP using CENP-A and CENP-C antibodies in MLL iKO cells. Consistent with our observations in IF, the level of CENP-A and CENP-C showed a reduction on α-satellite loci in cells devoid of MLL (Figs 7E, 7F, S6I and S6L). So far, we have demonstrated that MLL and SETD1A deposit H3K4me2 on the centromere and loss of these proteins decreases the cenRNA transcripts resulting in a decrease in centromere proteins like CENP-C and CENP-A. In order to establish that the loss of H3K4me2 is directly co-related to loss in CENP-A, we stained for CENP-A and H3K4me2 on chromatin fibers stretched from centromeres of cells treated with Control, MLL, or SETD1A siRNA (Fig 7G). Consistent with our IF and ChIP results, loss of MLL or SETD1A resulted in dramatic reduction of both H3K4me2 and CENP-A on the chromatin fibers (Fig 7G). This loss was dependent on the SET-domain activity of both proteins, as cells expressing MLLΔSET or SETD1AΔSET treated with MLL or SETD1A siRNA, respectively, could not rescue the endogenous levels of either H3K4me2 or CENP-A (S7F and S7G Fig). Altogether, our results indicate that MLL and SETD1A modulate the epigenetic state of the centromere by their histone methyltransferase/transcription activity to regulate the cellular machinery involved in CENP-A deposition.

## Discussion

The discovery of the active histone mark—H3K4me2—within the centrochromatin and the elegant demonstration of its crucial role in kinetochore maintenance using targeted-engineering of HAC, has firmly established the importance of this mark in kinetochore function. However, experimental evidence identifying which of the many histone lysine methylation enzymes deposits this mark on the centromere, was lacking. Despite differences in size, structure, interacting partners, and catalytic potential, the various members have been grouped under the MLL (KMT2) family banner by virtue of their SET domain. Even though each of these proteins is uniquely required during development, redundant roles of these enzymes are well known [26,80]. In this study, we have shown that the majority of MLL family members associate with the centromeres and regulate centromeric transcription. Our study here highlights that not only the H3K4me2 mark but the enzyme depositing it, also contributes to centromere stability.

### MLL regulates kinetochore assembly in multiple ways

Our results suggest that MLL affects the kinetochore assembly and maintenance in multiple ways. Not only is the presence of the SET domain in MLL required at the centromere to deposit the H3K4me2 mark, but it also co-activates centromeric transcription by its TA domain. Both these activities are pertinent in recruiting key proteins like CENP-C and HJURP to the centromere and therefore loading of CENP-A at the centromere. In fact, our results

indicate that loss of rapid transcription (by deletion of TAD) exhibits a pronounced effect on de novo CENP-A incorporation in our assay, even though the decrease in cenRNA transcripts by loss of both TAD and SET-domain deletion is comparable (Figs 1 and 7C and 7D). It is interesting to note that even though we observed the binding of MLL on centromere and its regulation of CENP-A deposition in multiple cell lines (Figs 2–3 and 6–7), we also observed cell-type specific MLL-mediated transcriptional regulation of proteins like CENP-B, which are important for centromere function. Further, we also note that MLL did not affect the levels of CCTT lncRNA, despite having a clear effect on the CENP-C/CENP-A loading, indicating that more than one pathway may exist for assembly of these proteins on the centromere.

Overall, loss of CENP-A is tolerated by the cell with chromosome segregation defects appearing after multiple rounds of cell division [4,6,15]. We have previously reported modest chromosome mis-alignment upon loss of SET or TA domain [33]. Indeed, mutation of the WDR5-interacting motif in MLL turned out to be the major cause of chromosome misalignment in our assays after 72 h of RNAi. However, we also believe that MLL participates in multiple pathways to regulate chromosome segregation. This statement is prompted by our earlier observation that about 90% of all MLL-depleted cells showed segregation defects but only 25% of these cells showed elongated phenotype that can be attributed to loss of Kinesin-like protein 2A (KIF2A) function [33]. Studies from the HAC model have revealed that kinetochores can function for several rounds of cell division without displaying any prominent defects even after the loss of H3K4me2, and with only 50% CENP-A on the centromere [4,6]. Our assays for nascent CENP-A deposition, though effective over multiple arrays, manifest loss of 50% CENP-A after 72 h of RNAi treatment. Remarkably, this loss is only 25% in MLLΔSET mutant indicating that these cells will take much more time to exhibit failure of kinetochore activity than can be addressed in our current assay. This could explain why we did not detect segregation defect as a primary phenotype of SET-domain deleted mutants in our earlier reports. Alternatively, the redundant functional activity of family members and/or alternate pathways could circumvent the loss of MLL. For example, yeast Set1A has been shown to regulate spindle assembly checkpoint through its interaction with mitotic arrest deficient 2 (Mad2) and regulate Ipl1-Aurora kinase by methylating outer-kinetochore protein Dam1 [34,81]; both processes ensure the proper segregation of chromosomes during mitosis. All in all, the regulatory relationship of the Set1A/MLL family members on the centromere is undeniable.

## Epigenetic landscape of human endogenous centromere: Differences from the HAC model

Our quest on understanding mitotic roles of MLL highlights the fact that loss of these proteins have a pleiotropic effect on cellular processes making it hard to attribute one defect to one process. Even though we performed RNA-seq experiments to gauge the impact of MLL's transcriptional function on its role in the deposition of CENP-A, we cannot discount the fact that our HMT(s) do regulate the expression of a large number of genes [31,45–48]. Further, transcription of alpha-satellite is an evolving field and all the factors/players affecting these processes are not known [2,16,61]. Therefore, all conclusions being made here, have to be considered in the background of large-scale transcription deregulation upon loss of these HMT(s). For such proteins, the use of HAC is an effective tool in teasing out mechanistic details of one process at a time on the centromere. On the flip side, till the cross-talk of multiple pathways is not appreciated, the cumulative impact of a histone modifier cannot be correctly gauged. For instance, in line with observations in HAC, we observed lower levels of active transcription marks (H3K4me2 and H3K9ac) and an increase in the inactive transcription mark (H3K9me3) on the centrochromatin in MLL iKOs. In contrast to the reports in

HAC [4,6], we observed an increase in the transcription elongation mark H3K36me2 in MLL iKO cells at the centromere. Without a doubt, H3K36me2 mark is primarily associated with active transcription [82–84]. In agreement, besides abovementioned reports, another study observed that loss of KDM2A, the H3K36me2 demethylase, is associated with higher α-satellite transcription [85]. However, different from reports in HAC, we also observed an accumulation of RNA Pol II $^{S2P}$ (and R-loops) in MLL iKOs (also see below). An increase in aberrant R-loops can lead to DNA double-strand breaks at several genomic locales and challenge centromere integrity [53,54,86]. Interestingly, H3K36me2 increases in cells with exaggerated DNA double-strand breaks [87–89], a phenomenon which has been reported on centromeres [54,57]. Hence, our data can be better explained considering the nature of H3K36me2 both as a transcription elongation mark and a DNA break repair factor, a possibility that needs further testing.

## MLLs regulate co-transcriptional R-loops at the centromere

Previous studies report a positive correlation between transcription, H3K4me2 mark and R-loop formation [38,90], and R-loops at centromere are no different [57]. Our data with SETD1A knock-down confers with this model and showed a decrease in H3K4me2 and R-loop formation. In contrast, the loss of MLL, despite showing decreased levels of H3K4me2 on the centromere, gave rise to an increase in R-loop formation. Even though, this MLL-RNAi-based R-loop formation was transcription dependent, we observed a concomitant increase in RNA Pol II$^{S2P}$ at the centromere depicting an aberrantly "paused" RNA Pol II in MLL iKO cells. Indeed, a stalled RNA pol II is associated with R-loop formation [58]. Loss of function of MLL has been reported to result in varied defects in RNA Pol II distribution [31,46,63]. While reduced RNA Pol II occupancy has been reported at some loci, an increase in RNA Pol II$^{S2P}$ and serine 5-phosphorylated RNA Pol II (RNA Pol II$^{S5P}$) forms have been reported at others [31,46]. A recent study shows an increase of RNA Pol II levels at transcription termination sites in MLL KO cells [63] indicating that abnormal distribution of Pol II can ensue following a loss of MLL.

Due to our inability to generate SETD1A knock-out cells, we had to make use of SETD1A shRNA knock-down to interrogate the status of RNA Pol II$^{S2P}$ in the absence of SETD1A on the centromere. Our data indicate that the total RNA Pol II as well as RNA Pol II$^{S2P}$ levels decreased upon loss of SETD1A. These observations are in contrast to studies conducted in yeast *Saccharomyces cerevisiae*, which show that deletion of the only H3K4 methyltransferase (*Set1A*) in the cell has little effect on the recruitment of RNA Pol II on constitutive euchromatin [83,84]. Another study in higher organisms reported that SET domain deleted mutant of SETD1A showed no compelling changes in RNA Pol II occupancy in mammalian cells [91,92]. These differences in RNA Pol II levels may arise specially at centrochromatin, which due to its epigenetic makeup presents an environment different from the rest of the chromatin [2]. Indeed, the centrochromatin is permissive to transcriptional elongation allowing limited activity of RNA Pol II [4,6,12]. Previously, it was reported that levels of RNA Pol II$^{S2P}$ fell progressively in the absence of H3K4me2 in HAC [4,6]. Whether RNA Pol II$^{S2P}$ levels decreased here as a cause or consequence of SETD1A depletion is hard to tell. In any case, we report a novel and contrasting difference between MLL and SETD1A in R-loop formation on the centromere. Our study also highlights the fact that despite being such well-studied co-activators of RNA Pol II, how different members of the MLL family regulate RNA Pol II, at various genomic loci, is still not clear.

R-loops reported at centromeres have been proposed to be beneficial [53,56,57] as well as detrimental [54,55] to centromere integrity. Recent reports suggest that CENP-A and Aurora

B Kinase can prevent the formation of opportunistic R-loops at centromeres in a cell-stage-specific manner [54,93]. Furthermore, we found that loss of both MLL and SETD1A severely impacts CENP-A loading at the centromeres. While the loss of MLL and SETD1A can trigger replication stress and DNA damage [78,94] that are associated with deleterious R-loops [86], how R-loop imbalance at centromeres challenges DNA integrity is still an open question. Altogether, our work raises an interesting hypothesis that MLL and SETD1A may regulate different classes of R-loops and thus impact centromere integrity.

### Implications of centromeric transcription regulation in MLL-rearranged leukemias

Growing body of evidence suggests perturbed centromeric and pericentromeric transcription in pathological conditions like cancer [2,95,96]. For example, in lung cancer and squamous cell carcinoma, dysregulation of centromeric transcription was observed accompanied by a global loss of repressive epigenetic marks [96]. Similarly, loss of chromatin regulatory proteins has been reported to induce centromeric transcription as a cause or consequence of oncogenesis [85,87]. Furthermore, a decrease in centromeric transcription is lethal to the cell as the absence of tumor suppressor Pbx-regulating protein-1 (Prep1) leads to an increase in repressive marks, resulting in centromere instability [97]. Additionally, kinetochore proteins like CENP-K and KNL-1 have been reported as fusion partners of MLL in leukemia [98]. Intriguingly, long noncoding (lnc) RNAs seem to play a crucial role in MLL-mediated gene regulation [71,99]. For example, HOTTIP is a well-studied lnc RNAs that interacts with MLL-WDR5 and regulates transcription of *HOXA*-gene cluster through looping of chromatin in normal cells, failure of which could trigger leukemogenesis in mice. Another study reports that lnc RNA UMLILO interacts with MLL-WDR5 and imparts trained immunity in mice [71]. Here, we found that MLL is regulating the expression of cenRNAs; however, further studies understanding the role of MLL-fusions in centromeric transcriptions still need to be undertaken.

## Methods

### Cell culture and stable cell line generation

U-2OS (human osteosarcoma), HEK-293 (human embryonic kidney), and IMR-90 tert (human lung fibroblast) cells were grown as monolayers in DMEM, supplemented with 10% (v/v) fetal bovine serum (FBS), 1% (v/v) GlutaMAX, and 100 U/ml penicillin-streptomycin. The cells were maintained at 37°C in a humidified atmosphere with 5% $CO_2$. All cell lines were authenticated by Lifecode Technologies Private Limited (India).

### Cloning and site-directed mutagenesis

U-2OS cells expressing MLL mutants have been described before [35] except MLLΔTAD that was generated in full-length MLL here by deletion of aa 2847–2855 using site-directed mutagenesis. Full-length SETD1A cDNA, gift from Robert Roeder [100], was cloned in Xho1 linearized pcDNA5/FRT-SFB vector [60]. We then generated siRNA resistant full-length SETD1A by introducing 7 silent mutations in the full-length construct using site-directed mutagenesis (see S1I Fig). The siRNA resistant full-length SFB-tagged SETD1A construct was further used to generate SETD1AΔSET (Δaa1407-1707) and SETD1A N1646A plasmids using site-directed mutagenesis. *MLL* sgRNA were cloned into a lentiGuide-Puro vector, gift from Feng Zhang [101] in the BsmB1 site. CENPA-SNAP-3xHA ORF, gift from Lars Jansen [79] was cloned into EcoR I and Kpn I linearized pcDNA 3.1-Puro vector or Hind III and Xho I linearized pcDNA FRT vector (Thermo Fischer Scientific).

## Generation of stable cell lines

Generation of MLL cell lines has been described previously [35]. SETD1A cell lines were generated by transfecting the cells with SETD1A constructs using polyethylenimine (PEI; Polysciences) as described earlier [60]. Plasmid-transfected cells were selected in the media supplemented with 200 μg/ml Hygromycin B (Thermo Fischer Scientific). Cells cultured from individual colonies were used for further experiments. To generate inducible knockouts for MLL, HEK-293 cells were transduced with Doxycycline inducible Cas9 expression vector pCW-Cas9, a gift from Eric Lander and David Sabatini [52] as lentiviral particles, and colonies stably expressing Cas9 protein were selected using 5 μg/ml Blasticidin (Thermo Fischer Scientific). These stable Cas9-expressing cells were then transduced with viral particles carrying MLL sgRNA. After transduction, *mll* knock-out colonies were selected with 2 μg/ml Puromycin (Gibco). Several colonies were screened for loss of MLL protein expression through western blot and finally, 2 clones (MLL iKO #11 and MLL iKO #20) were selected for further analysis. Cas9 expression (and therefore MLL knockout) was induced with 5 μg/ml Doxycycline (Sigma) treatment for 7 days where the medium was replenished after every 3 days. For CENP A-SNAP experiments, pcDNA FRT-CENP A-SNAP-3xHA was transfected into U-2OS and MLL mutant cells lines, and selected in media supplemented with 200 μg/ml Hygromycin B. Similarly, the pcDNA 3.1-Puro-CENP A-SNAP-3xHA construct was transfected into U-2OS and SETD1A mutant cell lines and selected using 4μg/ml Puromycin. Despite several attempts, we were unsuccessful in generating stable cell lines for pcDNA FRT-CENP A-SNAP-3xHA with MLLΔTAD. Therefore, pcDNA FRT-CENP-A-SNAP-3xHA vector was transiently transfected into MLLΔTAD cell line 12 h before siRNA transfection and assay was performed as depicted in Fig 7A.

## RNA interference

RNAi was performed with synthetic siRNA duplexes using Oligofectamine (Thermo Fischer Scientific) as previously described [33]. The sequence of siRNA targeting the firefly luciferase gene (used as Control) and various members of MLL family has been provided in S1 Methods [35]. Samples were collected 48 to 72 h after the first round of transfection as mentioned in the legends. For double knock down of MLL and SETD1A, the same amount of siRNA duplexes were used for each target as were used for knock down of a single target.

## Immunoblotting

Whole-cell protein extracts were prepared by lysing cells in 2X NETN buffer (200 mM NaCl, 40 mM Tris-Cl (pH 8.0), 1 mM EDTA, and 1% NP-40) supplemented with a freshly prepared protease inhibitor cocktail (Sigma) and boiled for 10 min. Equal amounts of protein extracts were resolved by SDS-PAGE and transferred to either PVDF or nitrocellulose membrane. Immunoblotting was performed with the following antibodies: MLL (A300-374A, Bethyl Labs); SETD1A (A300-288A, Bethyl Labs); CENP-A (2186S, Cell Signaling Technology); CENP-B (ab25734, Abcam), CENP-C (ab50974, Abcam), HJURP (80508S, Cell Signaling Technology), HA (H6908, Sigma), and α-tubulin (T5168, Sigma). After probing with relevant secondary antibodies, blots were developed using Amersham ECL substrate or digital imaging using LI-COR Biosciences as described [102].

## Immunofluorescence

Cells (U-2OS, MLL, and SETD1A mutants expressing cell lines) used for immunofluorescence were grown on coverslips. Cells were arrested with nocodazole (100 ng/ml) treatment for 12 to

16 h and released into fresh medium for 60 min (mitotic cells) or 90 min (early G1), before fixing. Cells were fixed with freshly prepared 1% paraformaldehyde for 10 min at room temperature followed by permeabilization with 0.2% Triton X-100/PBS for 10 min. For staining RNA-DNA hybrids, cells were fixed and permeabilized with 100% ice-cold methanol for 10 min followed by acetone for 1 min on ice. Following fixation, immunofluorescence staining protocol was followed as described earlier [33]. The samples were mounted in VECTASHIELD Mounting Medium (Vector laboratories-H1200) with 4′-6-diamidino-2-phenylindole (DAPI) to stain the DNA. Images were taken using a ZEISS LSM LSM 700 inverted confocal microscope with a 63×/1.4 oil immersion and quantified with either ZEN (ZEISS Efficient Navigation) or Image J software. Primary and secondary antibodies used for immunofluorescence were as follows: CENP-A (ab13939, Abcam), CENP-B (ab25734, Abcam), CENP-C (ab50974, Abcam), HJURP (80508S, Cell Signaling Technology), MLLC (A300-374A, Bethyl Labs); MLLN (A300-086A, Bethyl Labs); MLL2 (A130-173, Bethyl Labs); MLL3 (sc130173, Santa Cruz Technology); SETD1A (A300-288A, Bethyl Labs), SETD1B (A302-281A, Bethyl Labs), S9.6 (MABE1095, Sigma or ENH001, Kerafast), total RNA Polymerase II (sc-9001, Santa Cruz Technology), and RNA polymerase IIS2P (ab24758, ab252855, Abcam), Alexa Fluor 488 (A11029, A11034, Invitrogen), Alexa Fluor 594 (A11032, A11037, A21209, Invitrogen). For IF signal intensity quantification, Z-stack images with 0.5 μm step size were taken. To quantify centromeric signal of CENP-A, CENP-B, CENP-C, and HJURP, the signal intensity was calculated by manually placing a circle (of equal radius) around the centromere in maximum intensity projection images and the pixel value of each channel was calculated. For each channel, the background intensity was also calculated by placing another circle adjacent to the centromere signals, which was then subtracted from the respective IF channel value. To quantify colocalization of MLL and SETD1A with CENP-A in mitosis and early G1, single plane images were used. 3D view and Pearson correlation coefficient analyzes were performed using ZEN black software.

## Chromatin immunoprecipitation

ChIP was performed as described previously [58,60]. Briefly, approximately 80% confluent HEK-293, IMR-90 tert, MLL inducible knock-out cells were fixed with 1% formaldehyde for 10 min at room temperature to perform cross-linking and quenched with 250 mM glycine for 5 min (also known as X-ChIP). Cells were lysed and sonicated to shear chromatin to achieve approximately 200 to 500 bp fragments. However, to immunoprecipitate centromeric proteins (CENP-A and CENP-C), native, non-cross-linked chromatin immunoprecipitation (also known as N-ChIP) was performed using a modified protocol described earlier [103]. Briefly, cells were biochemically fractioned to get whole nuclei [104]. To fragment the chromatin, whole nuclei were resuspended in 100 μl MNase digestion buffer (15 mM NaCl, 10 mM Tris-Cl (pH 7.4), 60 mM KCl, and 1 mM CaCl2) with 4U of MNase (N3755, Sigma) incubated at 37°C for 10 min. The reaction was inhibited immediately by quick chilling on ice and by the addition of 100 μl MNase stop buffer (100 mM EDTA, 100 mM EGTA, 0.05% NP-40). Then, ChIP lysis buffer was added and incubated on ice for 15 to 30 min.

For both kinds of ChIP experiments, 1/10th of the fragmented chromatin was taken aside as input. The following antibodies were used for ChIP experiments: H3K4me2 (ab32356, Abcam); H3K36me2 (ab9049, Abcam); H3K9me3 (ab8898, Abcam); H3K9Ac (ab4441, Abcam), MLL (A300-374A, Bethyl Labs), SETD1A (A300-288A, Bethyl Labs), total RNA polymerase II (14958, Cell Signaling Technology or sc-9001, Santa Cruz Technology); RNA polymerase IIS2P (ab252855, Abcam); CENP-A (ab13939, Abcam), CENP-B (ab25734, Abcam), CENP-C (ab50974, Abcam), and IgG (12–370, Sigma). After incubation with primary

antibodies overnight at 4°C followed by the addition of Protein A or G Sepharose beads (GE Healthcare) for 2 to 3 h, immunoprecipitated material was washed with ChIP wash buffers. Immunoprecipitated and input DNA were subsequently purified using standard phenol, chloroform, and isoamyl alcohol extraction. The relative occupancy or percent input of the immunoprecipitated protein at each DNA locus was estimated by RT-qPCR as follows: $100 \times 2(Ct$ Input–Ct IP), where Ct Input and Ct IP are mean threshold cycles of RT-qPCR on DNA samples from input and specific immunoprecipitations, respectively. To measure fold over control, fold change over the ChIP values obtained in the control cells was used. The primer sequences are listed in S1 Methods.

## DNA:RNA immunoprecipitation

DNA DNA:RNA hybrids immunoprecipitation (DRIP) was performed as described earlier [58, 105] with the following modifications. Briefly, cells were lysed in 300 μl lysis buffer (100 mM NaCl, 10 mM Tris (pH 8.0), 25 mM EDTA (pH 8.0), 0.5% SDS) and sonicated using Diagenode Bioruptor (5 cycles of the 30s ON and 30s OFF at low intensity) and incubated with 100 μg/ml Proteinase K overnight at 37°C. Nucleic acids were extracted from phenol-chloroform extraction and resuspended in DNase/RNase-free water. Nucleic acids were fragmented using a restriction enzymes cocktail (50U each of EcoRI, BamHI, HindIII, and XhoI). Fragmented DNA served as inputs. About 2 to 5 μg of fragmented DNA was digested with 40U RNaseH (New England Biolabs) for at least 24 h at 37°C to serve as a negative control. After cleaning digested nucleic acids with phenol-chloroform extraction and re-suspended in DNase/RNase-free water, S9.6 antibody (MABE1095, Sigma) was added in a 1:1 ratio of nucleic acid: antibody in binding buffer (10 mM NaPO4 (pH 7.0), 140 mM NaCl, 0.05% Triton X-100) and incubated overnight at 4°C. Immunoprecipitated complexes were pull-down using Protein A Sepharose beads (GE Healthcare) were at 40°C for 2 h. Isolated complexes were washed thrice with ice-cold binding buffer and once with TE buffer for 2 min each, before elution (50 mM Tris (pH 8.0), 10 mM EDTA, 0.5% SDS, 5 μg proteinase k) for 30 min at 55°C. Nucleic acids were extracted using standard procedures. The relative occupancy or percent input of the immunoprecipitated DNA-RNA hybrids at each locus was estimated by RT-qPCR as follows: $100 \times 2$ (Ct Input–Ct IP), where Ct Input and Ct IP are mean threshold cycles of RT-qPCR on samples from input and specific immunoprecipitations, respectively. To measure fold over control, fold change over the DRIP values obtained in the control cells was used. The primer sequences are listed in S1 Methods.

## RNA isolation and qRT-PCR

Total RNA isolation and cDNA preparation was made as described earlier [33]. Briefly, total cellular RNA was isolated from MLL and SETD1A mutant cells or parent U-2OS cells using the TRIzol (Ambion) or using an RNA isolation kit (Zymo Research). To inhibit RNA polymerase II transcription, U-2OS cells were treated with the following inhibitors: 20 μm Triptolide (T3652, Sigma), 20 μm LDC000067 hydrochloride (SML2179, Sigma), and 20 μg α-amanitin (A2263, Sigma) for 4 h. For cDNA synthesis, the isolated RNA was treated with TURBO DNase (Thermo Fisher Scientific) for 30 min at 37°C to remove genomic DNA contamination and no-enzyme RNA amplification was done before cDNA synthesis to ensure that purified RNA did not have DNA contamination. cDNA was synthesized using SuperScript III Reverse Transcriptase kit (Thermo Fischer Scientific) according to the manufacturer's protocol. RT-qPCR was performed in either 7500 Real-Time PCR (Applied Biosystems), QuantStudio 5 Real-Time PCR System (Applied bioscience), or Bio-Rad (CFX-maestro) using

DyNAmo ColorFlash SYBR Green qPCR kit (Thermo Fisher Scientific). The transcript levels were quantified using 2-ΔΔ Ct [106]. The primer sequences are listed in S1 Methods.

## SNAP quench-pulse labeling of nascent CENPA

SNAP quench-pulse labeling was performed as described earlier [79] with the following modifications. Parent U-2OS, MLL, and SETD1A mutant cells stably expressing CENPA-SNAP-HA were seeded on coverslips and treated with Control, MLL, or SETD1A siRNA for 48 h. Cells were arrested with thymidine (2 μm) for 12 h followed by the treatment of 5 μm O6 –BG (BG-block) in complete growth media for 30 min at 37˚C to quench the SNAP activity. The Blocker was removed by washing cells twice with PBS, once with medium, and finally replenished with a fresh growth medium to ensure the complete removal of a blocker reagent. Cells were then arrested with nocodazole (100 ng/ml) for 12 h and released for 3 h followed by treatment of TMR-Star (2 μm, Covalys) for 30 min and stained using HA antibody (H6908, Sigma).

## Chromatin fiber assay

Extended chromatin fiber assay was performed as described [11,107] with following modification: Cells were grown on coverslips and treated with 75 mM KCl for 5 min. After hypotonic swelling, cells were quickly transferred into 1 ml of freshly prepared chromatin lysis buffer (25 mM Tris (pH 7.5), 500 mM NaCl, 400 mM Urea, 1% TX100) and incubated for 20 min at RT. After lysis, chromatin fibers were drawn by tilting the cover slip 80 degrees for 30 s, fixed immediately in 3.7% paraformaldehyde for 10 min, washed twice with PBS and blocked in 1% BSA for 1 h at RT. After blocking, the slides were processed for indirect immunofluorescence staining.

## Quantification and statistical analysis

GraphPad Prism 9.3 software was used to perform statistical analysis. Student *t* test and two-way ANOVA were performed as mentioned in the legends. In two-way ANOVA, significance is calculated against mean of control versus mean of test (or IgG). Error bars represent standard error of the mean (SEM) or standard deviation (SD) wherever mentioned in the legends. For exact number of cells and experiments, please refer to figure legends.

## RNA-Seq analysis

Total cellular RNA was isolated either from Control, MLL, or SETD1A siRNA-treated cells using TRIzol (Ambion). Libraries for RNA-Seq were prepared using NEB Ultra II RNA Directional Library kit. Paired-end sequencing (2 × 150 bp) was performed on Illumina NextSeq 2000 at the sequencing facility—National Genomics Core (NGC), CDFD. The raw sequencing read pairs were trimmed by removing adapters using TrimGalore (v 0.6.7) (https://github.com/FelixKrueger/TrimGalore). Trimmed sequencing reads were mapped to the human reference genome (GRCh38) using STAR (v. 2.7.10a) [108]. BAM files were indexed using SAM tools (v 1.13) [109]. Read counts quantification was done using feature Counts [110]. DESeq2, an R Bioconductor package was used to identify differentially expressed genes (DEGs) [111]. *P*-values were adjusted using Benjamini–Hochberg method, and the genes with adjusted *P*-value <0.05 and a fold change of 1.0 were identified as DEGs.

Pathway and gene ontology analysis was performed using DAVID (Database for Annotation, Visualization and Integrated Discovery), a web server for functional annotation and enrichment analyzes of gene lists [112]. Centromere, cell cycle, and transcription by RNA Pol II gene ontology data was downloaded from the Gene Ontology Annotation (GOA) database

[113]. All the results were analyzed using in-house shell, R, Perl, and Python scripts. All relevant data is available within the manuscript and Supporting Information files. RNA sequencing data associated with the manuscript have been deposited into the Gene Expression Omnibus database under the accession number GSE231942.

## Supporting information

**S1 Fig. RNAi-mediated down-regulation of MLL family members abrogates centromeric transcription.** (**A**) siRNA-mediated down-regulation of various MLLs was performed and the efficacy of siRNA treatment was determined by plotting the qRT-PCR analysis of respective transcript levels after total RNA extraction and cDNA synthesis. (**B–E**) Immunoblots of whole-cell lysate were prepared from cells treated with either control and MLL (**B**), MLL2 (**C**), SETD1A (**D**), or SETD1B (**E**) siRNA to analyze the respective protein levels. Blots were probed with α-MLLC (**B**), α-MLL2 (**C**), α-SETD1A (**D**), and α-SETD1B (**E**) and α-tubulin as shown. Molecular weight markers (in kDa) and relative quantification are shown as indicated. Uncropped blots provided in S1 Raw Images. (**F**) cDNA samples obtained after RNAi treatment of Control, MLL2, MLL3, and SETD1B (from **A**) were analyzed for cenRNA transcripts from D17Z1, D17Z1-B, and D17Z1-C α-satellite arrays of chromosome 17. (**G**) qRT-PCR analysis of α-satellite cenRNA from individual HOR of chromosome 17 and RNA Pol II regulated gene *PAX3*, after treatment with either control (DMSO), Triptolide (20 μm), CDK9 inhibitor (20 μm), α-amanitin (20 μg) for 4 h, is shown. (**A**, **F**, **G**) cDNA was synthesized from total RNA after rigorous DNase I treatment and amplified using qRT-PCR for indicated RNAs. Data from all samples were normalized to GAPDH mRNA levels from respective samples by using − ΔΔCT method and expression is shown relative to control siRNA-treated/ DMSO-treated cells from respective cell line/treatment (which is arbitrarily set to 1). Each experiment was performed at least 3, or more times except α-amanitin treatment (2 times). Error bars represent SD. *$P \leq 0.05$, **$P \leq 0.005$, ***$P \leq 0.0005$, ****$P \leq 0.0001$, ns: not significant, $P > 0.05$ (two-tailed Student's *t* test). (**H**) Schematic representation of recombinant MLL and SETD1A FLAG-epitope tagged mutants used in this study displaying different domains in these proteins. Full-length MLL (FL) and SET domain deleted MLL (Δaa3829–3969) U-2OS cell lines have been described before [35]. MLLΔTAD was generated in full-length MLL here by deletion of aa 2847–2855 using site-directed mutagenesis. U-2OS cells expressing full-length SETD1A (FL), SET domain deleted SETD1A (Δaa1407-1707), or point mutation inactivating SET domain (SETD1A N1646A mutant) were generated from siRNA resistant full-length SETD1A cDNA described in **I**. (**I**) siRNA resistant full-length SETD1A was generated by introducing 7 silent mutations (shown in red) at wobble positions between nucleotide +2916 to +2934 (Accession no NM_014712.3) in the full-length construct using site-directed mutagenesis. The siRNA (siRNA#1) sequence is underlined. CDK9i, CDK9 inhibitor; AT hook, AT-rich region; F, FLAG epitope tag; PHD, Plant homeodomain; Zn CXXC, Zinc-finger domain; Bromo, Bromodomain; RRM, RNA recognition motif; FYRN/C, Phenylalanine and tyrosine-rich region N-terminal/C-terminal; Win, WDR5 interacting motif. The raw data underlying parts (**A**) and (**F**, **G**) can be found in S1 Data.
(TIF)

**S2 Fig. MLLs bind to the human centromere repeats.** (**A**) Immunofluorescence staining (IF) of endogenous MLL (green) or SETD1A (green) with CENP-A (red) in U-2OS cells in interphase is shown. DNA was stained with DAPI (blue). The area in the white square is magnified and shown on the right for each image. (**B**) A 3D view of endogenous MLL (green) upper panel or SETD1A (green) lower panel. with CENP-A (red), is shown. The magnified area of the 3D model is shown on the right. (**C-D**) U-2OS cells were transfected with Control, MLL

(**C**), or SETD1A (**D**) siRNA to check for the specificity of MLL or SETD1A staining at the centromere. Cells were stained with endogenous MLL (green) or SETD1A (green) with CENP-C (red) or CENP-A (red), and DNA was stained with DAPI (blue) as indicated. (**E**) U-2OS cells, stained with Alexa Flour 488 and Alexa Flour 594, are shown. (**A–E**) Scale bar, 5 μm. (**F**) U-2OS cells were stained with MLL2 (green), MLL3 (green), or SETD1B(green) antibody along with centromeric marker CENP-A (red) as shown. The area in the white square is magnified and shown on the left for each image. Scale bar, 2 μm. (**G, I**) Immunoblot show MLL (**G**) and SETD1A (**I**) shRNA (#1 and #2) knockdown efficiency in treated HEK-293 cells. The blots were probed with α-MLL (**G**) or α-SETD1A (**I**), and α-α-tubulin antibody. Uncropped blots provided in S1 Raw Images. (**H, J**) Chromatin immunoprecipitation (ChIP) analyzes showing decrease in levels of MLL (**H**) and SETD1A (**J**) at centromeric α-satellite loci following treatment of either MLL shRNA or SETD1A shRNA in HEK-293 cells, and the result plotted as percent input enrichment, are shown. Each experiment was performed at least 3 or more times. Error bars represent SD. $*P \le 0.05$, $**P \le 0.005$, $***P \le 0.0005$, $****P \le 0.0001$, ns: not significant, $P > 0.05$ (two-tailed Student's $t$ test). (**K**) CENP-B occupancy, detected by ChIP, is shown. Data from 3 or more independent ChIP experiments is shown. Error bars represent SD. $*P \le 0.05$, $**P \le 0.005$, ns: not significant, $P > 0.05$ (two-way ANOVA with Šídák multiple comparison test). α-sat, α-satellite. The raw data underlying parts (**H**) and (**J, K**) can be found in S1 Data.
(TIF)

**S3 Fig. Loss of MLL affects the epigenetic landscape of the centromeres.** (**A**) Cell cycle profile obtained from flow cytometry analysis of *MLL* KO (iKO #11 and #20) cells after 7 days of Doxycycline treatment is shown. Representative graph (from 3 independent experiments) is shown with values obtained for sample shown here. Parental (or Cas9-expressing) cells are used as Control. The raw data underlying this part can be found here (http://flowrepository.org/id/FR-FCM-Z6AH). (**B–F**) ChIP-analyzes with MLL (**B**), H3K4me2 (**C**), H3K9ac (**D**), H3K9me3 (**E**), and H3K36me2 (**F**) antibodies in *MLL* iKO cells (#20) are shown. Data were normalized against the ChIP values obtained in parental (or Cas9-expressing) cells, which are used as Control. Data from 3 or more independent ChIP experiments are plotted. Error bars represent SD. $*P \le 0.05$, $**P \le 0.005$, $***P \le 0.0005$, $****P \le 0.0001$, ns: not significant, $P > 0.05$ (two-way ANOVA with Šídák multiple comparison test). α-sat, α-satellite. The raw data underlying parts (**B–F**) can be found in S1 Data.
(TIF)

**S4 Fig. Disparate impact of MLL and SETD1A on centromeric R-loops.** (**A**) Representative images for the data represented in Fig 4A show nuclear R-loops, 48 h after MLL and SETD1A and MLL+SETD1A siRNA treatment in U-2OS cells. The cells were stained using the S9.6 (green) antibody and DAPI (blue). Each nucleus is outlined in white. For transcription inhibition, cells were treated with 20 μm Triptolide or DMSO (Control) for 4 h. Scale bar, 10 μm. (**B**) DRIP analysis in MLL RNAi-treated HEK-293 cells, with respective RNase H controls, is shown. Data are presented as percent input enrichment. Data from 3 or more independent DRIP experiments are plotted. (**C, D**) DRIP analysis in HEK-293 cells, with asynchronous or cells synchronized in mitosis, is shown. Data are presented as percent input enrichment. Data from 3 or more independent DRIP experiments are plotted. Data from **D** is replotted in **C** for Control cells to highlight the changes observed in mitosis. Error bars represent SD. $**P \le 0.005$, $***P \le 0.0005$, ns: not significant, $P > 0.05$ (two-way ANOVA with Šídák multiple comparison test). (**E, F**) Immunofluorescence staining of Total RNA Pol II (**E**, green) or RNA Pol II$^{S2P}$ (**F**, green) and CENP-A (red) in mitotic cells following treatment with either Control, MLL, or SETD1A siRNA, are shown. Scale bar, 5 μm. (**G–I**) ChIP-analysis of RNA

Pol II (**G**) and RNA Pol II$^{S2P}$ (**H**) in *MLL* knock out (iKO #11) and H3K4me2 (**I**) in SETD1A shRNA treated HEK-293 cells is shown. Data were normalized against the ChIP values obtained in Control samples and presented as fold change over control. Data from 3 or more independent experiments were plotted. Error bars represent SD. *$P \le 0.05$, **$P \le 0.005$, ****$P \le 0.0001$, ns: not significant, $P > 0.05$ (two-way ANOVA with Šídák multiple comparison test). Ctrl, control; si, siRNA; Asyn, asynchronous. The raw data underlying parts (**B–D**) and (**G–I**) can be found in S1 Data.
(TIF)

**S5 Fig. Transcriptional effects of MLL and SETD1A down-regulation on centromere genes.** (**A**, **B**) U-2OS cells were either treated with Control, MLL (**A**), or SETD1A (**B**) siRNA, and immunoblots of whole-cell lysate showing MLL (**A**), SETD1A (**B**), and respective tubulin level are shown. Uncropped blots provided in S1 Raw Images. Number of genes up- or down-regulated significantly ($\log2FC > 1$ or $\log2FC < -1$ and padj $< 0.05$, Benjamini–Hochberg method) in MLL (**A**) or SETD1A (**B**) siRNA-treated cells were listed (also see S1 Table). (**C**, **D**) GO enrichment analysis of differentially regulated transcripts were identified in MLL (**C**) and SETD1A (**D**) siRNA-treated cells. Biological processes of GO terms were ranked based on the adjusted *p*-value obtained from online web server DAVID. Ten most significant enriched GO terms are presented. The color represents padj value and the diameter of the circle size indicates no of genes in that category. (**E**, **F**) Venn diagram showing the overlap between the gene ontology annotations related to cell cycle, centromere, and transcription by RNA Pol II among DEGs in MLL-KD (**E**) and SETD1A-KD (**F**) (also see S2 and S3 Tables). (**G**) SETD1A siRNA-treated cDNA samples were analyzed for CENP-H transcript levels as shown. (**H**) cDNA samples obtained after RNAi treatment of Control, SETD1A, and MLL, were analyzed for CCTT lncRNA transcripts levels. (**G**, **H**) *$P \le 0.05$, **$P \le 0.005$, ns: not significant, $P > 0.05$ (two-tailed Student's *t* test). SET1, SETD1A; up/down, up-regulated/down-regulated genes; GO, Gene Ontology; DEGs, differentially expressed genes; CCTT, CENP-C targeting transcript. The raw data underlying parts (**G**, **H**) can be found in S1 Data.
(TIF)

**S6 Fig. MLL and SETD1A down-regulation does not influence centromeric protein levels.** (**A–D**) Immunoblot shows CENP-C (**A**), CENP-B (**B**), HJURP (**C**), and CENP-A (**D**) protein levels in Control or MLL siRNA-treated U-2OS cells. (**E–H**) Immunoblot shows CENP-C (**E**), CENP-B (**F**), HJURP (**G**), and CENP-A (**H**) protein levels in Control or SETD1A siRNA-treated U2OS cells. (**I–L**) Immunoblot analysis of CENP-C (**I**), CENP-B (**J**), HJURP (**K**), and CENP-A (**L**) protein levels in *MLL* HEK-293 KOs (iKO #11 and #20) cell lines are shown. Blots were probed with respective antibodies as indicated. Molecular weight markers (in kDa) are shown on the left and quantitative analysis of respective protein levels from 2 independent experiments is shown on the right. Each bar represents mean SD. **$P \le 0.005$, ns: not significant, $P > 0.05$. (**A–H**, two-tailed Student's *t* test; **I–L**, two-way ANOVA with Šídák multiple comparison test). Uncropped blots provided in S1 Raw Images. The raw data underlying parts (**A–L**) can be found in S1 Data.
(TIF)

**S7 Fig. MLL and SETD1A facilitate recruitment of nascent CENPA at centromeres.** (**A**) Quantification of HJURP fluorescence intensity following depletion of Control, MLL, SETD1A, or MLL+SETD1A. Each data point represents a single centromere. Quantification of HJURP fluorescence images shown in Fig **6G**) The error bar represents SEM $\ge 250$ centromeres quantified from 10 early G1 cell pairs, (*n* = 3 experiments). ****$P \le 0.0001$, *$P \le 0.05$, ns: not significant, $P > 0.05$ (Ordinary one-way ANOVA with Tukey's multiple comparisons

test). (**B**) Quantification of centromeric fluorescence intensity of ectopically expressed total CENP-A (stained using α-HA antibody) in parent U-2OS cells (—) or cell line stably expressing MLL full length (FL) or MLLΔTAD or MLLΔSET upon MLL siRNA treatment. (**C**) Quantification of centromeric fluorescence intensity of total CENP-A in parent U-2OS cells (—) or cell line stably expressing siRNA resistant SETD1A full length (FL) or SETD1AΔSET (here, N1646A mutant was used) upon SETD1A siRNA. Each data point represents a single centromere; $n \geq 300$ quantified from 10 early G1 cell pairs, ($n = 2$ experiments). ****$P \leq 0.001$, ns: not significant, $P > 0.05$ (Mann–Whitney two-tailed unpaired test). (**D, E**) Stably expressing CENP-A SNAP-3xHA cells were either transfected with Control, MLL (**D**), or SETD1A (**E**) siRNA, and collected after 72 h for whole-cell lysate preparation. CENP-A-SNAP-3xHA protein was detected using an α-HA antibody. Uncropped blots provided in S1 Raw Images. Quantitative analysis of relative total protein levels are shown (lower panel). Each bar represents mean SD of 2 biological replicates. *$P \leq 0.05$, ns: not significant. (**F, G**) Immunofluorescence staining (IF) of chromatin fiber prepared from cell line stably expressing MLLΔSET (**F**) or SETD1AΔSET (**G,** here N1646A mutant was used). Stably cell line cells were transfected either with Control, MLL siRNA (**F**), or SETD1A (**G**) siRNA and the chromatin fibers were extracted and stained with endogenous CENP-A (green) and H3K4me3 (red), and DNA stained with DAPI (blue). (**F**) Frequency of H3K4me2 co-localizing with CENP-A was found to be 12/15 in Control. This dropped to 5/28 upon MLL siRNA treatment. (**G**) Frequency of H3K4me2 co-localizing with CENP-A was found to be 8/8 in Control and 0/6 upon SETD1A siRNA treatment. Scale bar, 10 μm. The raw data underlying parts (**A–E**) can be found in S1 Data. (TIF)

**S1 Methods. shRNA transfections and sequences of primers used in this study.** (DOCX)

**S1 Table. Differentially expressed genes upon MLL and SETD1A knockdown.** (XLS)

**S2 Table. Gene ontology enrichment analysis.** List of differentially expressed genes upon MLL and SETD1A knockdown for GO term—Cell Cycle. (XLS)

**S3 Table. Gene ontology enrichment analysis.** List of differentially expressed genes upon MLL and SETD1A knockdown for GO term—transcription by RNA Pol II. (XLS)

**S4 Table. Analysis of important genes involved in CENPA deposition.** Selected genes involved in CENPA deposition were analyzed upon MLL and SETD1A knockdown and results were compared to data obtained by RNA-Seq analysis shown in S1 Table. Raw data points along with adjusted *p* values represent qRT PCR validation of same genes. (XLSX)

**S1 Data. Underlying data for Figs 1–7.** (XLSX)

**S1 Raw Images. Original uncropped blots used in this study.** (PDF)

## Acknowledgments

We thank R. Roeder, F. Zhang, L. Jansen, S. Huang, E. Lander, and D. Sabatini for cDNA constructs. We would like to thank A. Karole for cloning MLL and SET1A shRNA and sgRNA

constructs, J. Thakur and Z.U. Zargar for their help with initial experiments, and S.E.F facility for technical support. We also thank I. Cheeseman, A. Desai, M.D. Blower, W.C. Earnshaw, and N. Kochanova for suggestions and discussions. KKM is recipient of Junior and Senior Research Fellowships of the Department of Biotechnology (DBT), India toward the pursuit of a PhD degree of the Manipal University. KAL and PDK are recipients of Junior and Senior Research Fellowships of the Council of Scientific and Industrial Research (CSIR), India toward the pursuit of a PhD degree of the Regional Centre for Biotechnology (RCB) and Manipal University respectively.

## Author Contributions

**Conceptualization:** Shweta Tyagi.

**Data curation:** Deepshika Pulimamidi.

**Formal analysis:** Kausika Kumar Malik, Sreerama Chaitanya Sridhara, Deepshika Pulimamidi.

**Funding acquisition:** Shweta Tyagi.

**Investigation:** Kausika Kumar Malik, Sreerama Chaitanya Sridhara, Kaisar Ahmad Lone, Payal Deepakbhai Katariya.

**Methodology:** Kausika Kumar Malik, Sreerama Chaitanya Sridhara, Kaisar Ahmad Lone, Payal Deepakbhai Katariya, Deepshika Pulimamidi.

**Project administration:** Shweta Tyagi.

**Supervision:** Shweta Tyagi.

**Validation:** Kausika Kumar Malik.

**Writing – original draft:** Kausika Kumar Malik, Sreerama Chaitanya Sridhara, Payal Deepakbhai Katariya, Shweta Tyagi.

**Writing – review & editing:** Shweta Tyagi.

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
