## [Editor Report · Decision Letter 0]

2 Aug 2022

Dear Dr Tyagi, 

Thank you for submitting your Review Commons manuscript entitled "MLL family members regulate H3K4 methylation to ensure CENP-A assembly at human centromeres." for consideration as a Research Article by PLOS Biology. Please accept my apologies for the delay in getting back to you as we consulted with an academic editor about your submission.

Your manuscript has now been evaluated by the PLOS Biology editorial staff, as well as by an academic editor with relevant expertise, and I am writing to let you know that we would like to pursue your manuscript further and invite you to submit a revised version of the manuscript.

However, before we can invite a revision, we need you to complete your submission by providing the metadata that is required for full assessment. To this end, please login to Editorial Manager where you will find the paper in the 'Submissions Needing Revisions' folder on your homepage. Please click 'Revise Submission' from the Action Links and complete all additional questions in the submission questionnaire.

To provide the metadata for your submission, please Login to Editorial Manager (https://www.editorialmanager.com/pbiology) within two working days, i.e. by Aug 04 2022 11:59PM.

Kind regards,

Richard

Richard Hodge, PhD

Associate Editor, PLOS Biology

rhodge@plos.org

PLOS

---

## [Editor Report · Decision Letter 1]

8 Aug 2022

Dear Dr Tyagi,

Thank you very much for submitting your manuscript " MLL family members regulate H3K4 methylation to ensure CENP-A assembly at human centromeres" for consideration as a Research Article at PLOS Biology. As you know, your manuscript and plan of revision have been evaluated by the PLOS Biology editors and by an Academic Editor with relevant expertise. 

Based on your responses to the reviews from Reviews Commons, we would welcome re-submission of a revised version that takes into account the reviewers' comments. In addition, for the revised version to be a strong candidate for publication, the Academic Editor would like you to consider potential indirect explanations for the role of MLL and SETD1A in centromere transcription and centromere biology. Please see the comments below from the Academic Editor regarding this point.

We cannot make any decision about publication until we have seen the revised manuscript and your response to the reviewers' comments at Review Commons. Your revised manuscript is also likely to be sent for further evaluation by the original reviewers.

We expect to receive your revised manuscript within 3 months. Please email us (plosbiology@plos.org) if you have any questions or concerns, or would like to request an extension. At this stage, your manuscript remains formally under active consideration at our journal; please notify us by email if you do not intend to submit a revision so that we may end consideration of the manuscript at PLOS Biology.

**IMPORTANT - SUBMITTING YOUR REVISION**

*Re-submission Checklist*

*Published Peer Review*

*PLOS Data Policy*

*Blot and Gel Data Policy*

Sincerely,

Richard

Richard Hodge, PhD

Associate Editor, PLOS Biology

rhodge@plos.org

Comments from the Academic Editor:

This paper implicates MLL (KMT2) and SetD1A in affecting various aspects of centromere biology. Given the diverse roles for these proteins in the gene expression programs across the cell, a major challenge for this paper is determining whether the phenotypes observed following depletion of these proteins reflect a direct role for these proteins at centromeres or whether these effects are indirect through altering gene expression more broadly. The localization of these proteins to centromeres does not exclude the possibility that the phenotypes resulting from their depletion can be explained by gene expression changes. This is a critical point that the author's need to address directly both through experiments and changes to the text.

For example, prior work has suggested that centromere proteins (CENP-A, CENP-B, etc), ATR, 53BP1, cohesin, RNA Polymerase I, the nucleolus, and MANY other factors also influence centromere transcription. Thus, changes in the expression of any of these factors, their regulators, or many others have the potential to alter centromere transcription indirectly. The authors conduct Western blots and cite published gene expression data to suggest that there are not substantial changes in a selected number of centromere components (although I note that the published data they cite at least suggests the possibility that CENP-A expression is altered). However, the data provided does not rule out these alternative explanations. In an ideal world, it would be valuable for the authors to provide robust RNA-seq data to evaluate the gene expression changes that occur in their knockdown conditions and in their cell types. They could also test some of these other possibilities more directly. In addition, even with these other controls, it is important that the authors be extremely cautious with their wording in the abstract, Results, and Discussion to highlight the multiple potential explanations and not to definitively state the conclusion that this reflects a direct role for MLL at centromeres. In addition to the experimental changes, this paper should include a clear acknowledgement of this potential alternative explanation and describe the limitations of their existing study and data.

---

## [Decision Letter · Decision Letter 2]

18 Apr 2023

Dear Dr Tyagi,

Thank you for your patience while we considered your revised manuscript "MLL family members regulate H3K4 methylation to ensure CENP-A assembly at human centromeres" for publication as a Research Article at PLOS Biology. Please accept my apologies for the delays that you have experienced during this round of the peer review process. This revised version of your manuscript has been evaluated by the PLOS Biology editors, the Academic Editor and three of the original reviewers at Review Commons.

In light of the reviews, which you will find at the end of this email, we are pleased to offer you the opportunity to address the comments from the reviewers in a revision that we anticipate should not take you very long. After discussions with the Academic Editor, we will not make the inclusion of additional experimental data a requirement for publication, but we do ask that the comments are fully addressed by caveating the limitations of the work in the manuscript text and discussing possible alternative explanations for the data. We will then assess your revised manuscript and your response to the reviewers' comments with our Academic Editor aiming to avoid further rounds of peer-review, although might need to consult with the reviewers, depending on the nature of the revisions.

In addition, I would be grateful if you could please address the following data and other policy-related requests that I have provided below (A-H):

(A) We would like to suggest the following modification to the title:

“MLL methyltransferases regulate H3K4 methylation to ensure CENP-A assembly at human centromeres”

(B) You may be aware of the PLOS Data Policy, which requires that all data be made available without restriction: http://journals.plos.org/plosbiology/s/data-availability. For more information, please also see this editorial: http://dx.doi.org/10.1371/journal.pbio.1001797

-Supplementary files (e.g., excel). Please ensure that all data files are uploaded as 'Supporting Information' and are invariably referred to (in the manuscript, figure legends, and the Description field when uploading your files) using the following format verbatim: S1 Data, S2 Data, etc. Multiple panels of a single or even several figures can be included as multiple sheets in one excel file that is saved using exactly the following convention: S1_Data.xlsx (using an underscore).

-Deposition in a publicly available repository. Please also provide the accession code or a reviewer link so that we may view your data before publication.

Figure 1A-B, 1D-E, 2C-H, 3B-F, 4A-G, 5C-F, 6C-F, 6H, 7C-F, S1A, S1F-G, S2H, S2J-K, S3B-F, S4B-D, S4G-I, S5G-H, S6A-L, S7A-C, S7D-E

(C) Please deposit the RNA-seq data in a public repository, such as the GEO database. Please provide the accession number of the deposition in the Data Availability Statement in the online submission form and ensure that the data is made publicly available.

(D) Please deposit the FCS files for the Flow Cytometry data (Figures S3A) in the FlowRepository database (https://flowrepository.org/). As before, please provide the accession number in the Data Availability Statement and ensure the data is made publicly available.

(E) Please also ensure that each of the relevant figure legends in your manuscript include information on *WHERE THE UNDERLYING DATA CAN BE FOUND*, and ensure your supplemental data file/s has a legend.

(F) We require the original, uncropped and minimally adjusted images supporting all blot and gel results reported in the following figures:

Figure 3A, S1B-E, S2G, S2I, S5A-B, S6A-L, S7D-E

We will require these files before a manuscript can be accepted so please prepare and upload them now. Please carefully read our guidelines for how to prepare and upload this data: https://journals.plos.org/plosbiology/s/figures#loc-blot-and-gel-reporting-requirements

(G) Please note that per journal policy, we do not allow the mention of "unpublished data” or other references to data that is not publicly available or contained within this manuscript (see top of page 23). Please either remove mention of these data or provide figures presenting the results and the data underlying the figure(s).

(H) Please ensure that your Data Statement in the submission system accurately describes where your data can be found and is in final format, as it will be published as written there.

We expect to receive your revised manuscript within three weeks. 

*Published Peer Review History*

*Press*

Kind regards,

Richard

Richard Hodge, PhD

Associate Editor, PLOS Biology

rhodge@plos.org

Reviewer remarks:

Reviewer #1: The authors have satisfactorily addressed all concerns raised.

Reviewer #3: This is a revised manuscript from Malik and colleagues examining the role of MLL and SetD1A in centromere specification and transcription. Overall, the revisions have improved the manuscript, and most of the issues that were raised previously have been addressed. I think the weakness of the manuscript is that the data show that MLL and SetD1A influence centromeric transcripts, but the experiments do not demonstrate that the centromeric transcripts are the mechanism by which CENP-A and HJUPR recruitment are altered in MLL and SetD1A knockdown. But I will stipulate that this may be difficult to directly demonstrate, and I do not think they have been sufficiently proven in the literature. 

It is very interesting that the data suggest that SetD1A and MLL may be functioning differently, as SetD1A alters CENP-H and CCTT expression, both of which influence CENP-A deposition (see Okada 2006). The manuscript incorrectly states that CENP-H does not influence CENP-A deposition, however, Okada et al 2006 showed otherwise. The data that show SetD1A + MLL knockdown have a more drastic phenotype is excellent to see in the manuscript and solves of the big issues I had with the original manuscript, which was why do Set1DA and MLL both show phenotypes. However, it should be noted that despite the statement on page 19, "Our observations indicate that both MLL and SETD1A contribute to CENP- A recruitment synergistically." there is no data for synergy, which would require additional analysis. However, they could very well be additive, which is consistent with altering multiple pathways. 

One last point is that the analysis of the transcriptional changes is insufficient to dispel the concern that MLL is altering transcription of upstream regulators of CENP-A deposition. Besides HJURP, CENP-C and CENP-A there are many other genes that could negatively alter CENP-A deposition. At the very least, the transcriptional effects of SetD1A and MLL should be shown for Mis18a, Mis18b, Mis18BP1, CENP-I and PLK1. There are others suggested in the literature as well, and the authors should make an effort to pick these out of the RT PCR data to show whether they are changed.

Reviewer #4: Major comment

By the addition of control and/or double knockdown experiments in each revised figure, datasets are now comparable and easier to understand than before. In addition, the title was changed to focus solely on functional link between endogenous centromere H3K4me2 transferase and nascent CENP-A assembly mechanism. Data on the involvement of SET domain in the CENP-A assembly mechanism is also added, and supported conclusion of the title. The manuscript had been improved.

Minor comments

Since the double knockdown of MLL and SETD1A synthetically reduced nascent CENP-A assembly (Figure 6H), unique properties of both MLL and SETD1A, such as the different PolII or R-loop regulation manner, might also be involved in nascent CENP-A assembly. If the functional link between the PolII or R-loop regulation and nascent CENP-A assembly is investigated, the Figure 4 will fit the manuscript more tightly.

In the Figure 7G, how can the authors identify which fiber is centromere? And, how frequent can the authors observe such the events per centromere fiber? If the authors add the results of fiber imaging, number or frequency of the events should be shown.

---

## [Editor Report · Decision Letter 3]

12 May 2023

Dear Shweta,

On behalf of my colleagues and the Academic Editor, Iain Cheeseman, I am pleased to say that we can accept your manuscript for publication, provided you address any remaining formatting and reporting issues. These will be detailed in an email you should receive within 2-3 business days from our colleagues in the journal operations team; no action is required from you until then. Please note that we will not be able to formally accept your manuscript and schedule it for publication until you have completed any requested changes.

PRESS

Kind regards,

Richard 

Richard Hodge, PhD

Associate Editor, PLOS Biology

rhodge@plos.org

PLOS
